# Proimmunogenic impact of MEK inhibition synergizes with agonist anti-CD40 immunostimulatory antibodies in tumor therapy

Daniel Baumann[1,2], Tanja Hägele[1], Julian Mochayedi[1], Jennifer Drebant[1], Caroline Vent[1,2], Sven Blobner [1], Julia Han Noll [1], Irena Nickel[1], Corinna Schumacher[1], Sophie Luise Boos[1,3], Aline Sophie Daniel[1], Susann Wendler[1,2], Michael Volkmar[1,2], Oliver Strobel[2] & Rienk Offringa[1,2✉]

Cancer types with lower mutational load and a non-permissive tumor microenvironment are intrinsically resistant to immune checkpoint blockade. While the combination of cytostatic drugs and immunostimulatory antibodies constitutes an attractive concept for overcoming this refractoriness, suppression of immune cell function by cytostatic drugs may limit therapeutic efficacy. Here we show that targeted inhibition of mitogen-activated protein kinase (MAPK) kinase (MEK) does not impair dendritic cell-mediated T cell priming and activation. Accordingly, combining MEK inhibitors (MEKi) with agonist antibodies (Abs) targeting the immunostimulatory CD40 receptor results in potent synergistic antitumor efficacy. Detailed analysis of the mechanism of action of MEKi shows that this drug exerts multiple pro-immunogenic effects, including the suppression of M2-type macrophages, myeloid derived suppressor cells and T-regulatory cells. The combination of MEK inhibition with agonist anti-CD40 Ab is therefore a promising therapeutic concept, especially for the treatment of mutant Kras-driven tumors such as pancreatic ductal adenocarcinoma.

[1] Department of Molecular Oncology of Gastrointestinal Tumors, German Cancer Research Center Heidelberg, Heidelberg, Baden-Wuerttemberg 69120, Germany. [2] Department of Surgery, Heidelberg University Hospital, Heidelberg, Baden-Wuerttemberg 69120, Germany. [3]Present address: Department of Oncogenic signaling pathways of colorectal/pancreatic cancer, Ludwig-Maximilians-Universitaet, Munich, Bavaria 80539, Germany. ✉email: r.offringa@dkfz.de

Cancer immunotherapy is beginning to realize its potential in the clinic with immune checkpoint blockade (ICB), eliciting durable responses in patients with immunogenic cancers such as melanoma and lung cancer[1]. Nevertheless, sub-groups of patients with these indications do not respond to ICB, and the same applies to patients with other cancer types. A major hurdle in this respect is the lower intrinsic immunogenicity of ICB-resistant tumors, which is at least partly related to the number of somatic mutations encoding potential T cell epitopes[2]. The immunogenicity of many tumors is further decreased by loss of surface MHC class I expression[3], suppression of the antigen-processing machinery[4], and defects in interferon signaling[5]. Last but not least, immunosuppressive mechanisms in the tumor microenvironment (TME) other than the PD-L1/PD-1 and CTLA-4 pathways may render checkpoint blockade ineffective, such as inhibitory myeloid cell types and CD4+ T-regulatory cells[6–8].

In order to build on the first promising results of ICB, there is a clear need to explore additional drugs and treatment regimens in clinical trials. Agonist immunostimulatory antibodies (IS-Abs) targeting activatory receptors on immune cells are a potential alternative for immune checkpoint inhibitors and have demonstrated notable results in preclinical models[9]. An attractive aspect of agonist anti-CD40 Abs, which make these complementary to checkpoint inhibitors, is the capacity to enhance the priming of T cell responses through the activation of dendritic cells (DCs)[10]. Anti-CD40 Abs may therefore be able to turn 'cold' tumors into 'hot' tumors[11]. Furthermore, anti-CD40 Abs were found to make tumors more permissive to immune responses by modifying the immune-suppressive myeloid infiltrate in tumors[10]. In view of the DC-activating capacities of anti-CD40 Ab, preclinical, and clinical testing of these Ab primarily focuses on combination with cytostatic oncology drugs. The underlying idea of this treatment concept is that tumor cell killing results in 'immunogenic tumor cell death', involving tumor antigen release and cell death-related pro-inflammatory signals, causing the recruitment of antigen presenting cells (APCs) including DCs into the tumor, followed by the uptake and processing of tumor antigens into MHC[12]. In this context, stimulation of DCs through their CD40 receptor provides an 'adjuvant' signal, causing the antigen-loaded DCs to migrate to the tumor-draining lymph nodes[13–15], as well as to convert them into fully mature DCs expressing high levels of MHC-restricted tumor antigen in the context of multiple T cell co-stimulatory signals[9].

The first described example of a synergistic combination of an anti-CD40 Ab with a cytostatic drug concerned the chemotherapeutic drug gemcitabine (GEM)[11,16]. GEM treatment is relatively mild and rarely induces lymphopenia, making it more suitable for combination with immunostimulatory drugs than highly cytotoxic chemotherapeutic regimens. As GEM is used in the standard of care treatment of pancreatic ductal adenocarcinoma (PDA), and this cancer indication represents an urgent unmet medical need[17], this finding triggered the testing of this drug combination in both preclinical models for PDA and in clinical trials[11]. The interest in immunotherapeutic regimens for PDA was further boosted by the finding that this tumor, in contrast to long-standing views is infiltrated with tumor-reactive T cells and even contained ectopic lymphoid structures[18–20].

The dominant driver of PDA tumor growth is mutated KRAS[21], making these tumors highly dependent on the mitogen-activated protein (MAPK) kinase (MEK) pathway. Nevertheless, clinical responses of PDA to pharmacological MEK inhibition have been disappointing[22,23]. In the present study, we demonstrate that combination of anti-CD40 Ab with MEK inhibition results in a superior therapeutic regimen. Analysis of the mechanism of action on the basis of immune-related

parameters shows that this therapeutic efficacy is owing to the fact that MEK inhibitors exert strong antitumor cytotoxicity along with pro-immunogenic effects on tumor and TME that are complementary to the immunostimulatory action of anti-CD40 Ab.

## Results

**Cytostatic small molecule inhibitors for immuno-oncology.** In initial in vitro screens, we validated the biological activity of multiple small molecule inhibitors (SMi) targeting the MEK/ERK and PI3K/AKT pathways using well-defined human tumor lines in which these pathways are known to be activated. Benchmarking of the IC50 values against data available from literature led to the selection of MEK inhibitors (MEKi) GDC-0973 (cobimetinib), GDC-0623, as well as PI3K inhibitors (PI3Ki) GDC-0941 (pictilisib) and BAY 84-1236 (copanlisib) for further testing (Supplementary Fig. 1A–D). These compounds were subsequently evaluated for their in vitro cytostatic impact against two C57BL/6-based syngeneic tumor models commonly used for tumor immunology studies: MC-38 and B16-OVA (Fig. 1a, Supplementary Fig. 1E). Given our specific interest in PDA, we also tested drug impact on a newly generated, C57BL/6-based tumor line PDA30364, which was established from a tumor that arose in a genetically engineered PDA model driven by KRAS-G12D and P53-R172H. In accordance with the presence of these driver mutations, this cell line was sensitive to all MEK and PI3K inhibitors tested (Fig. 1a, Supplementary Fig. 1E). We focused our further experiments on MEKi GDC-0623, because we obtained most consistent in vitro efficacy data for this compound for the three tumor cell lines of interest (Fig. 1b).

GDC-0623 was reported to have superior efficacy in KRAS-driven tumors, owing to its capacity to block MEK feedback phosphorylation by wild-type RAF[24]. This was confirmed by the in vitro sensitivity of primary patient-derived PDA cell lines to this drug (Supplementary Fig. 1F) and is in line with the detection of activating KRAS mutations in these cell lines (Supplementary Data 1). Mutational analysis of the three mouse tumor lines revealed that, as expected, mutated KRAS is a key driver mutation in PDA30364. B16-OVA and MC-38 do not harbor KRAS mutations, but instead carry a number of other potential driver mutations that could stimulate the MEK/ERK pathway, either directly or through cross-talk between signaling pathways (Supplementary Data 1). Notably, all three tumor cell lines express elevated pERK levels that are profoundly inhibited through incubation with GDC-0623 (Fig. 1c, Supplementary Fig. 2). Nevertheless, the impact of MEK inhibition on cell viability is most prominent in PDA30364, in line with the pivotal role of mutated KRAS in this cell line (Fig. 1a). Detailed analysis of the balance between cell death and G0/G1 arrest, as induced in the three tumor cell lines by GDC-0623, confirmed that the degree of cell death was highest for PDA30364, whereas this drug induced a blend of cell death and stasis in B16-OVA and MC-38 (Fig. 1d–e).

**MEK inhibition does not impair DC and T cell function in vivo.** The MEK/ERK pathway was reported to be essential for T cell priming and effector function[25–30]. We initially examined GDC-0623 for potential suppressive impact on T cell function in in vitro experiments with antigen-stimulated TCR-transgenic, chicken ovalbumin (OVA)-specific CD8+ OT-I T cells. The resulting data pointed at profound inhibition of T cell proliferation and cytokine production at tumor-effective concentrations (Supplementary Fig. 3). Similar observations were made for the other three SMi (Supplementary Fig. 4). In contrast, testing of GDC-0623 in in vivo immunization experiments with OT-I

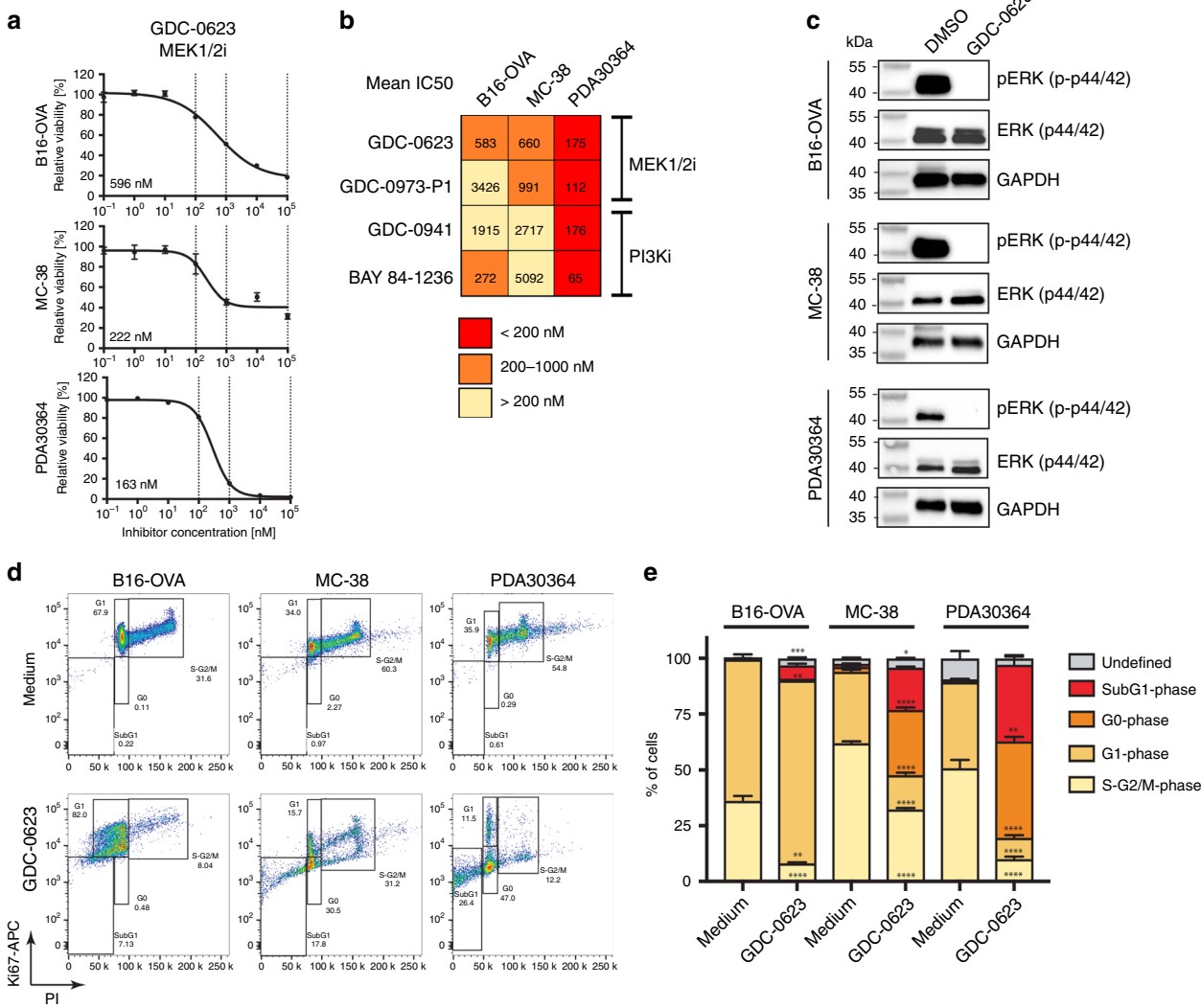

**Fig. 1 In vitro potency of MEK and PI3K inhibitors against syngeneic tumor cell lines.** ATP-based assessment of tumor cell viability in presence of indicated concentrations of small molecule inhibitors. Relative cell viability normalized to medium control. Inhibitory concentration (IC) 50 values calculated with 4-parameter logistic curve fit with bottom constraint >0. Mean ± s.e.m, $n = 3$. **a** Representative dose–response curves of murine tumor cell lines. **b** Mean IC50 values for indicated murine tumor cell lines of three independent experiments. Each experiment comprises three biological replicates. Shading indicates level of inhibitor potency (red: < 200 nM = sensitive; orange: 200–1000 nM = moderate; light yellow: > 1000 nM = not sensitive). **c** Western Blot analyses of indicated murine tumor cell lines treated with 0.5 μM GDC-0623 for 1 h. Representative data shown from two technical replicates. **d, e** Ki67/PI-based cell cycle analyses of tumor cells treated with 1 μM GDC-0623 and respective quantification. Mean ± s.e.m, $n = 3$. Unpaired student $t$ test (medium vs. GDC-0623 for each cell cycle phase; FDR ($Q = 1\%$), two-stage step-up method of Benjamini, Krieger and Yuketieli). Significance levels are indicated by asterisks (*$p \leq 0.05$; **$p \leq 0.01$; *** $p \leq 0.001$; ****$p \leq 0.0001$).

T cells revealed that antigen-induced T cell priming and proliferation were only slightly affected by drug doses known to suppress tumor growth in xenograft models[24] (Fig. 2a–b). These in vivo experiments involved systemic immunization of mice with OVA protein in combination with a systemic administration of agonist anti-CD40 Ab (anti-mouse CD40 Ab 3/23 mouse IgG1). In this setting, effective activation of the OT-I T cells depends on the co-administration of the anti-CD40 Ab (Supplementary Fig. 5A), which induces the activation of DCs and thereby mediates the immunogenic presentation of the OVA-derived epitope SIINFEKL to T cells[31,32].

Further evaluation of MEK inhibition in in vivo cytotoxicity assays demonstrated that also T cell-mediated killing was only minimally affected (Fig. 2c). Our data therefore demonstrate that daily dosing of GDC-0623 at 10 or 30 mg kg⁻¹ does not impair the function DCs with respect to the uptake, processing, and presentation of protein antigen, the subsequent priming and

clonal expansion of T cells, and also not the cytotoxic effector function of these T cells. Similar results were obtained in the in vivo immunization assays for MEKi GDC-0973 and PI3Ki GDC-0941 (Supplementary Fig. 5B, C). In contrast, profound inhibition of the T cell response was observed when OVA-immunized mice were treated with the chemotherapeutic drugs GEM or temozolomide (TEM) (Fig. 2d). Owing to their greater selectivity, the targeted SMi drugs tested may therefore be more suitable for combining with immunostimulatory drugs.

**Antitumor efficacy of MEKi with agonist anti-CD40 Ab.** As the aforementioned immunization experiments demonstrated that daily administration of GDC-0623 did not block the pro-immunogenic impact of agonist anti-CD40 Ab, we proceeded by testing the therapeutic efficacy of this Ab in combination with MEK inhibition in our tumor models. In line with our

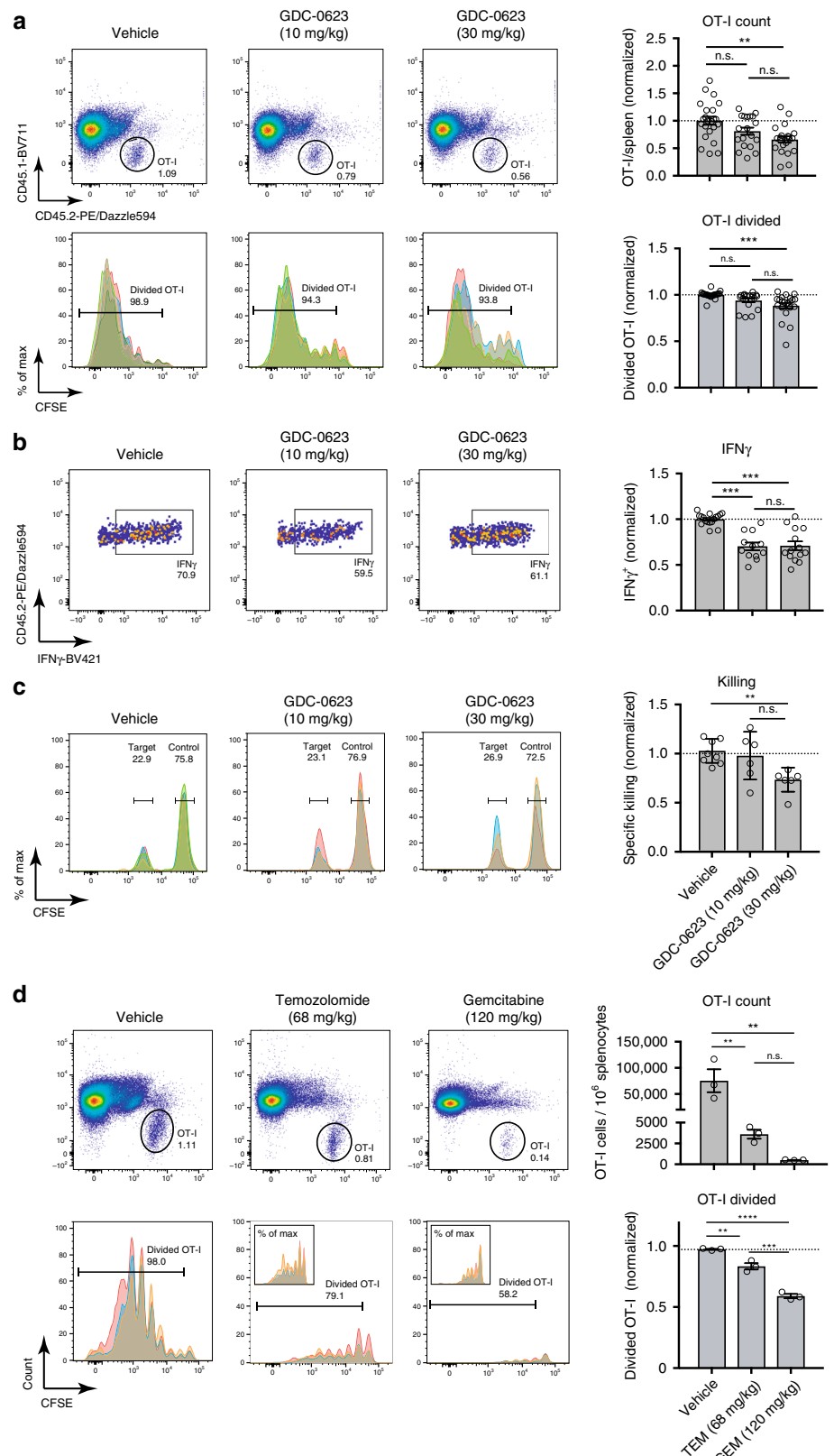

considerations, application of GDC-0623 with anti-CD40 Ab in the MC-38 and B16-OVA models showed striking synergy between these drugs, resulting in significant suppression of B16-OVA tumor growth and tumor stasis in the MC-38 model (Fig. 3a/b; Supplementary Fig. 6A). The impact of either drug alone was very limited in these tumor models. For the MEKi this

may be explained by the finding that also in the in vitro experiments GDC-0623 does not induce complete killing, but mostly stasis, in these tumor lines (Fig. 1a, d–e). As expected on basis of the in vitro data, mutant KRAS-driven PDA30364 tumors showed a much more profound in vivo response to MEKi alone (Fig. 3c; Supplementary Fig. 6A). The strong in vivo impact of

**Fig. 2 Chemotherapeutic agents, but not MEK inhibitors, impair T cell proliferation and activation in vivo. a** C57BL/6-Ly5.1 (CD45.1+) mice infused with CFSE-labeled OT-I T cells (CD45.2+) were treated for 6 days with the indicated doses of MEK inhibitor GDC-0623 or vehicle. Mice were immunized with OVA protein admixed with anti-CD40 Ab. After 3 days, splenocytes were analyzed. Upper row: dot plots of gated OT-I T cells. Lower row: CFSE dilution of gated OT-I T cells; differentially colored histograms represent data from individual mice. Bar charts to the right: cumulative data from six experiments (total mice: vehicle $n = 23$, GDC-0623 (10 mg kg$^{-1}$) $n = 19$, GDC-0623 (30 mg kg$^{-1}$) $n = 21$), showing quantification of OT-I T cell numbers and divided fraction. One-way ANOVA with post hoc Tukey's test. Significance levels are indicated by asterisks (*$p \leq 0.05$; **$p \leq 0.01$; ***$p \leq 0.001$; ****$p \leq 0.0001$). **b** Intracellular IFNγ levels of OT-I T cells after ex vivo restimulation with SIINFEKL peptide. Bar charts to the right: cumulative data of four experiments (total $n \geq 12$ mice per group). **c** CD45.1+ mice infused with naive CD45.2+ OT-I T cells were immunized with OVA protein admixed with anti-CD40 Ab. Three days later, mice were infused with CFSE-labeled peptide pulsed target cells (CD45.2+ splenocytes pulsed with 10 µg ml$^{-1}$ SIINFEKL; 0.3 µM CFSE) admixed with control cells (CD45.2+ splenocytes pulsed with 10 µg ml$^{-1}$ P53 control peptide AIYKKSQHM; 5 µM CFSE); spleens were analyzed for target cell killing after 24 h. GDC-0623 was administered at indicated doses for 3 days after immunization. Bar charts to the right: cumulative data of two experiments (vehicle $n = 8$, GDC-0623 (10 mg kg$^{-1}$) $n = 6$, GDC-0623 (30 mg kg$^{-1}$) $n = 6$) with specific killing normalized to the T cell only group. **d** As described in **a**, mice were treated on days 0 and 3 with indicated doses of temozolomide (TEM), gemcitabine (GEM), or vehicle. Upper row: dot plots of gated OT-I T cells. Lower row: CFSE dilution of gated OT-I T cells. Bar charts to the right: quantification of OT-I T cell numbers and divided fraction (three mice/group).

GDC-0623 on PDA30364 furthermore confirmed that the 30 mg/kg dose, which did not impair T cell priming in the OVA/ OT-I T cell immunization experiments, is relevant for tumor therapy. Importantly, also in the PDA30364 model, anti-CD40 Ab significantly enhances the antitumor impact of MEKi (Fig. 3c; Supplementary Fig. 6A). Although single-agent MEKi treatment results in transient tumor stasis in this model, prolonged stasis and tumor rejection was only achieved in MEKi/CD40 Ab-treated mice (Fig. 3d).

Others recently reported that MEK inhibition promoted T cell and antitumor activity in combination with PD-L1/PD-1 checkpoint blockade in the AT3ova, MMTV-neu, and CT26 tumor models[30,33]. Evaluation of anti-PD-1 Ab treatment in our PDA30364 model showed no single-agent activity, in contrast to what is reported for the aforementioned models. Furthermore, the combination of MEKi with anti-PD-1 Abs suppressed PDA30364 tumor outgrowth with comparable efficiency as MEKi single-agent treatment. In this tumor model, only the MEKi/ CD40 Ab combination resulted in long-term control of tumor outgrowth (Fig. 3e).

**Impact MEKi/CD40 Ab treatment on T cell infiltrate.** The mechanism of action of the aforementioned drugs and combination treatments was further analyzed at the level of the cellular immune response. Depletion of T cells abolished the efficacy of the MEKi/CD40 Ab treatment, underscoring the importance of the T cell immunity in this respect (Fig. 3f, Supplementary Fig. 6B). In line with this finding, a prominent feature of the MEKi/CD40 Ab regimen is the increase in CD8+/CD4+ Treg ratio, as observed by flow cytometry in all three tumor models (Fig. 4a/b). Analysis of single-agent-treated tumors revealed that the increase in CD8+ T cells is primarily driven by anti-CD40 Ab (Fig. 4c), whereas both anti-CD40 Ab and MEKi exert a suppressive effect on Tregs (Fig. 4d). The stimulation of the effector T cell responses by the combination treatment is further illustrated by the increased capacity of freshly isolated tumor-infiltrating T cells to produce effector cytokines (Fig. 4e, Supplementary Fig. 6C). These latter data vividly illustrate the anergic state of the T cells in the untreated tumors and, furthermore, that GDC-0623 synergizes with anti-CD40 Ab in unleashing T cell effector function. Notably, there is also a modest increase in cytokine production by T cells isolated from PDA30364 tumors treated with MEKi only (Fig. 4e).

In comparison, treatment of PDA30364 tumors with anti-PD-1 Abs elicited only a minor increase in the CD8+ T cell fraction and CD8+/Treg ratio (Supplementary Fig. 7A). A more profound impact on the CD8+/Treg ratio is seen under MEKi/ PD-1 Ab treatment, but this is most likely owing to the aforementioned

suppression of CD4+ Tregs by GDC-0632. Furthermore, the cytokine production by freshly isolated tumor-infiltrating T cells is equally enhanced for MEKi/CD40 Ab and MEKi/PD-1-treated tumors (Supplementary Fig. 7B). Altogether, these data indicate that the MEKi/CD40 Ab regimen is more effective, because it increases both the activity of tumor-infiltrating T cells and the magnitude of the T cell response (see further below).

**Impact MEKi on myeloid cell infiltrate.** We found that MEKi also have a profound impact on the myeloid immune infiltrate. We focused our analyses on the MC-38 and PDA30364 tumors, because the B16-OVA tumor model used by us was found to contain relatively few myeloid cells (Supplementary Fig. 8). As shown in Fig. 5a, untreated MC-38 and PDA30364 tumors are dominated by CD206+ M2-like macrophages. MEKi/CD40 Ab combination treatment results in a striking overall reduction in the number of macrophages, whereas the macrophages left in the treated tumors are mostly of the iNOS+ M1-type (Fig. 5a). Dissection of the action of each of the drugs through analysis of the single drug-treated groups showed that anti-CD40 Ab can enhance the M1 phenotype while suppressing the M2 phenotype. Interestingly, MEKi alone elicited an even stronger suppression of the M2 CD206+ phenotype (Fig. 5a). Together, this explains how MEKi/CD40 Ab treatment resulted in a positive M1/M2 ratio in the TME. Notably, a similar shift in the M1/M2 ratio was not observed in PDA30364 tumors treated with the MEKi/PD-1 Ab combination (Supplementary Fig. 7C), which could further explain the superiority of the MEKi/CD40 Ab regimen in this model.

We examined the impact of MEKi on macrophages in greater detail in mouse bone marrow-derived M1 and M2 cultures that were obtained by incubating macrophage precursors in the presence of IFNγ/LPS or IL-4, respectively (Fig. 5b; Supplementary Fig. 9a–c). Addition of GDC-0623 to these cultures showed that M2-polarized macrophages are much more sensitive to this drug than M1 macrophages at relevant concentrations (Fig. 5c). Similar results were obtained with other MEKi, including several clinically tested compounds (Supplementary Fig. 9D).

Although macrophages dominate the myeloid compartment in MC-38 tumors, PDA30364 tumors predominantly contained CD11b+Gr1+ granulocytic myeloid cells, also referred to MDSCs (Fig. 5d). Analysis of PDA30364 tumors subjected to the different treatments revealed that MEKi, but not anti-CD40 or anti-PD-1 Ab, strongly reduced granulocytic myeloid cell numbers in the tumors, in particular Ly6C+Ly6G+ cells, whereas sparing Ly6G-Ly6C+ monocytic myeloid cells (Fig. 5e; Supplementary Fig. 7D). Accordingly, MDSC differentiated from bone marrow monocytes using GM-CSF and IL-6 are highly sensitive

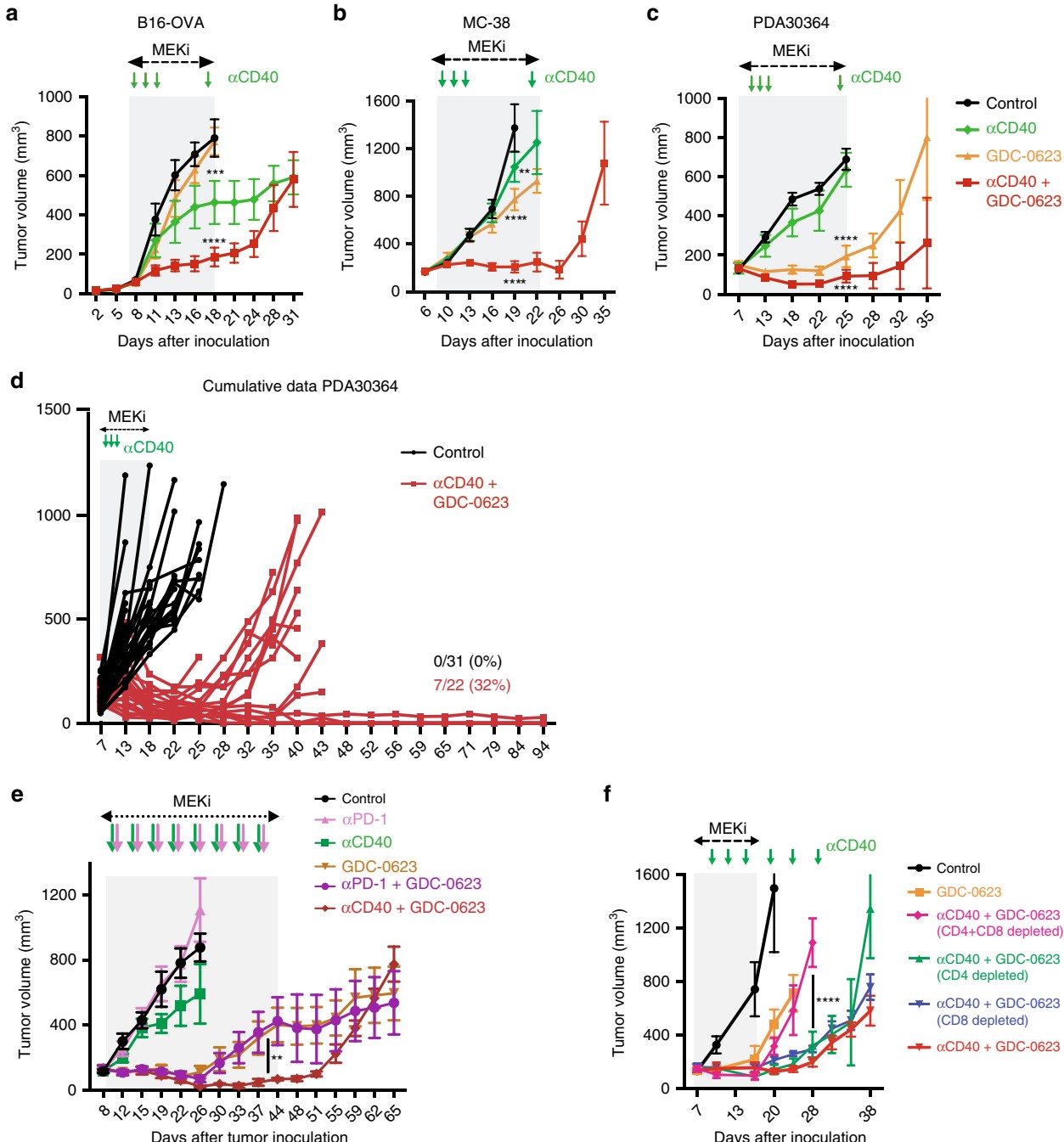

**Fig. 3 Combination of MEK inhibition with agonist anti-CD40 Ab mediates control of established tumors. a–c** Outgrowth of B16-OVA, MC-38, and PDA30364 tumors upon treatment with MEKi GDC-0623 and/or anti-CD40 Ab. Mice were treated daily with 30 mg kg$^{-1}$ GDC-0623 or vehicle for ~2 weeks (gray rectangle). Anti-CD40 or control Ab were administered on treatment days 3, 5, 7 as well as 1 day prior to biomarker analyses (green arrows). Each group consisted of at least six mice ($n_{B16-OVA} = 10$, $n_{MC38} = 10$, $n_{PDA30364} = 10$). Four mice per group were killed at end of treatment for biomarker analyses. Two-way ANOVA with post hoc Dunnett's test; treatment groups vs. control. Significance levels are indicated by asterisks (*$p \leq 0.05$; **$p \leq 0.01$; ***$p \leq 0.001$; ****$p \leq 0.0001$). **d** Cumulative data of several experiments with the PDA30364 tumor model. **e** PDA30364 tumor growth in mice treated with monotherapies (anti-CD40 Ab, anti-PD-1 Ab, or GDC-0623) and combination therapies (anti-CD40 Ab + GDC-0623 and anti-PD-1 Ab +GDC-0623). Mice were treated daily with 30 mg kg$^{-1}$ GDC-0623 or vehicle for ~5 weeks. Anti-CD40, anti-PD-1, and control Ab were administered twice weekly (arrows). Each group consisted of at least nine mice ($n_{Control} = 9$, $n_{CD40} = 9$, $n_{GDC-0623} = 12$, $n_{CD40+GDC-0623} = 10$, $n_{PD-1} = 10$, $n_{PD1-GDC-0623} = 11$). Four mice per group were killed 2 weeks after treatment start for biomarker analyses. **f** Impact of T cell depletion on efficacy of GDC-0623 and/or anti-CD40 Ab treatment against PDA30364 tumors. Where indicated, groups received depletory anti-CD4, anti-CD8 or anti-CD4+ anti-CD8 Abs twice per week. Each group consisted of at least five mice.

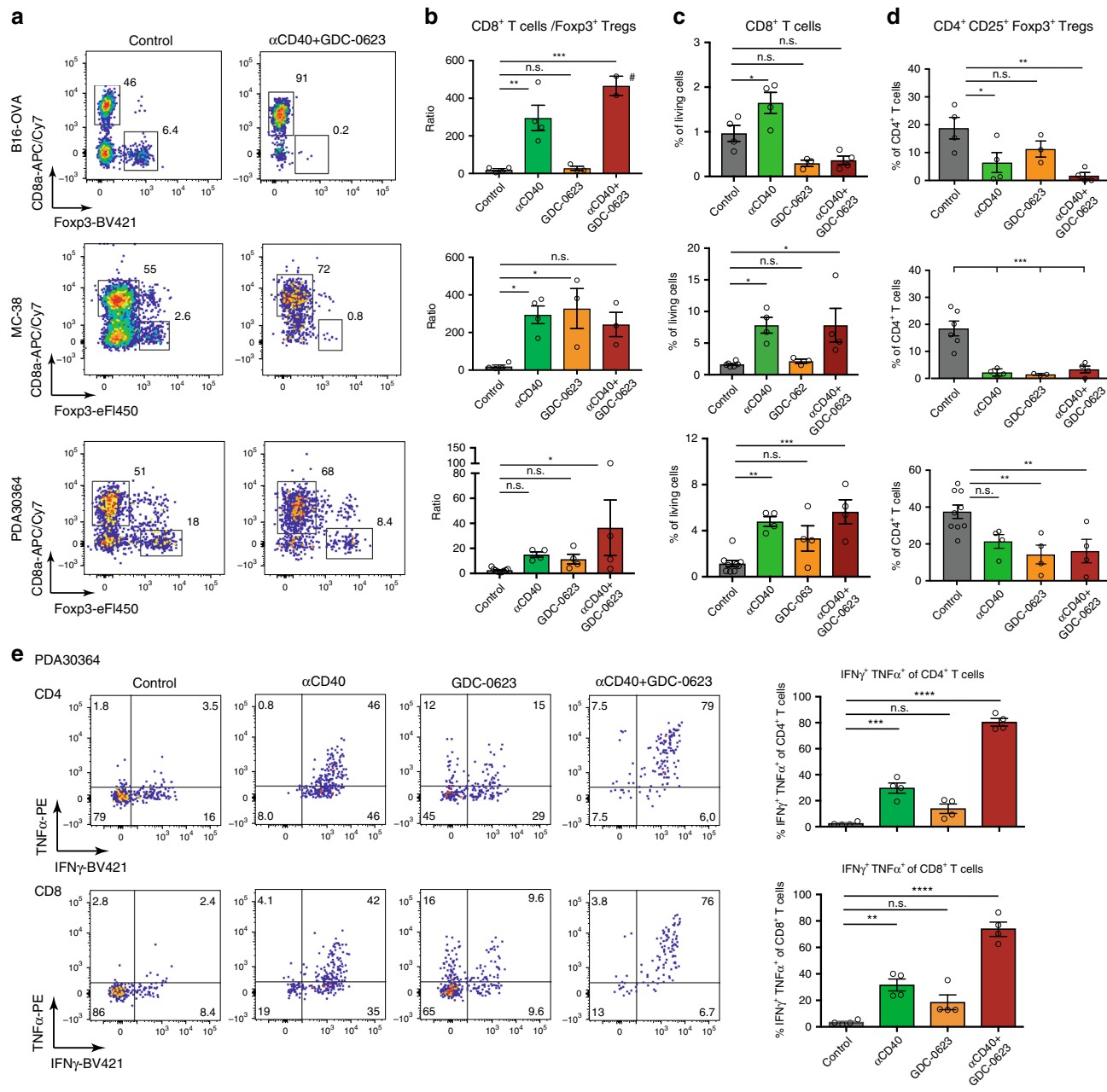

**Fig. 4 MEKi/anti-CD40 Ab combination therapy increases CD8/Treg ratio and effector function of tumor-infiltrating T cells. a** Flow cytometric analysis of the impact of MEKi GDC-0623 and/or anti-CD40 Ab treatment on the T cell infiltrate in B16-OVA, MC-38, and PDA30364 tumors, as performed on four mice per treatment group of the experiments shown in Fig. 3a–c. Representative dot plots are shown. Cells were pre-gated on living, CD45+, CD3+, Thy-1.2+ cells. **b–d** Quantitation of CD8+ T cell to CD4+ CD25+ Foxp3+ regulatory T cell ratio, regulatory T cell frequencies among CD4+ T cells, and CD8+ T cell frequencies among living cells. #Two animals without Tregs. One-way ANOVA with post hoc Dunnett's test (treatment groups vs. control). Mean ± s.e.m, n = 4. **e** TILs isolated from PDA30364 tumors were restimulated ex vivo with PMA/Ionomycin in presence of GolgiPlug and stained for intracellular IFNγ and TNFα. Mean ± s.e.m., n = 4. One-way ANOVA with post hoc Tukey's test. Significance levels are indicated by asterisks (*$p \leq 0.05$; **$p \leq 0.01$; ***$p \leq 0.001$; ****$p \leq 0.0001$).

to in vitro MEKi treatment (Fig. 5f), to the same extent as M2 macrophages (Fig. 5c).

**Analysis of drug mechanism of action by tumor transcriptomics.** To obtain an unbiased view on the mechanism of action of MEKi and anti-CD40 Ab in our tumor models, we performed transcriptome analysis of tumors isolated from mice undergoing treatment. Differential gene expression analysis between control and MEKi/CD40 Ab combination treatment

groups, followed by ingenuity pathway analysis (IPA), led to the identification of several distinct gene signatures associated with drug action. These signatures were most prominently observed in PDA30364 tumors, in line with the greatest degree of therapeutic impact of the treatment in this model. The first of these signatures, as identified by focusing on downregulated genes, reflects the suppression of cell cycle and biosynthesis-related pathways (Fig. 6a–b). Suppression of these pathways is clearly associated with the antitumor impact of MEKi when dosed either as single-

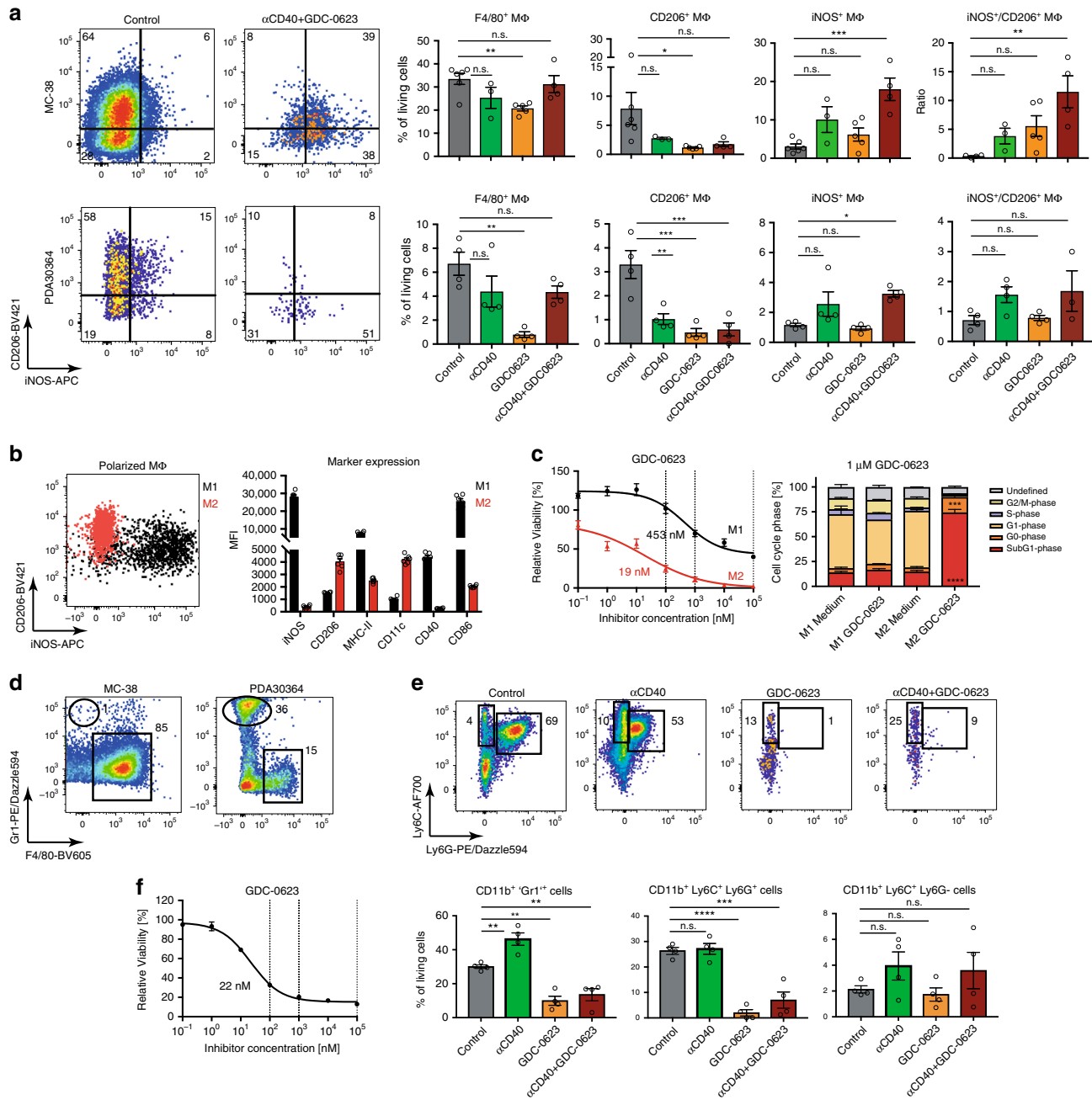

**Fig. 5 MEKi suppress M2-like macrophages and MDSCs. a** Flow cytometric analysis of the impact of MEKi GDC-0623 and/or anti-CD40 Ab treatment on the macrophage infiltrate in MC-38 and PDA30364 tumors, as performed on four mice per treatment group of the experiments shown in Fig. 3 b–c. Representative dot plots are shown. Cells were pre-gated on living CD45+, CD11b+, F4/80+ cells. One-way ANOVA with post hoc Dunnetts's test (treatment groups vs. control). Significance levels are indicated by asterisks (*$p \leq 0.05$; **$p \leq 0.01$; ***$p \leq 0.001$; ****$p \leq 0.0001$). **b** Polarization of mouse bone marrow-derived macrophage cultures towards M1- or M2-like phenotype, as assessed by flow cytometry marker analysis. Mean ± s.e.m, $n = 6$. **c** Cell viability of polarized macrophages after 3-day treatment with GDC-0623. Left panel: representative dose–response curves. Relative IC50 values were calculated by four-parameter logistic curve fit with a bottom constraint > 0. Mean ± s.e.m, $n = 3$. Right panel: Ki67/PI-based cell cycle analyses of tumor cells treated with 1 μM GDC-0623 and respective quantification. Mean ± s.e.m, $n = 3$. Unpaired student $t$ test (medium vs. GDC-0623 for each cell cycle phase; FDR ($Q = 1\%$), two-stage step-up method of Benjamini, Krieger and Yuketieli). **d** Representative dot plots of myeloid cell populations in MC-38 and PDA30364 tumors. **e** Representative dot plots of 'Gr1', Ly6C+Ly6G+, and Ly6C+ Ly6G− myeloid cell populations in PDA30364 tumors. Cells were pre-gated on living CD45+, CD11b+ cells; quantitation normalized to all living cells. Mean ± s.e.m, $n = 4$. One-way ANOVA with post hoc Dunnett's test (treatment groups vs. control). **F** Cell viability of bone marrow-derived MDSC cultures after 3-day treatment with GDC-0623; assay performed as described above. Mean ± s.e.m, $n = 3$.

agent or in conjunction with anti-CD40 Ab (Fig. 3c). This notion is supported by the fact that these pathways are also suppressed in MEKi treated in vitro PDA30364 cell cultures (Fig. 6c).

Evaluation of the tumor transcriptome in the B16-OVA and MC-38 tumors showed that also in these models suppression of

the cell cycle and biosynthesis signature correlated with antitumor treatment efficacy (Supplementary Fig. 10A). For example, strong suppression of these pathways was only seen in the MEKi/CD40-treated tumors, whereas no significant suppression of these pathways was seen in the B16-OVA

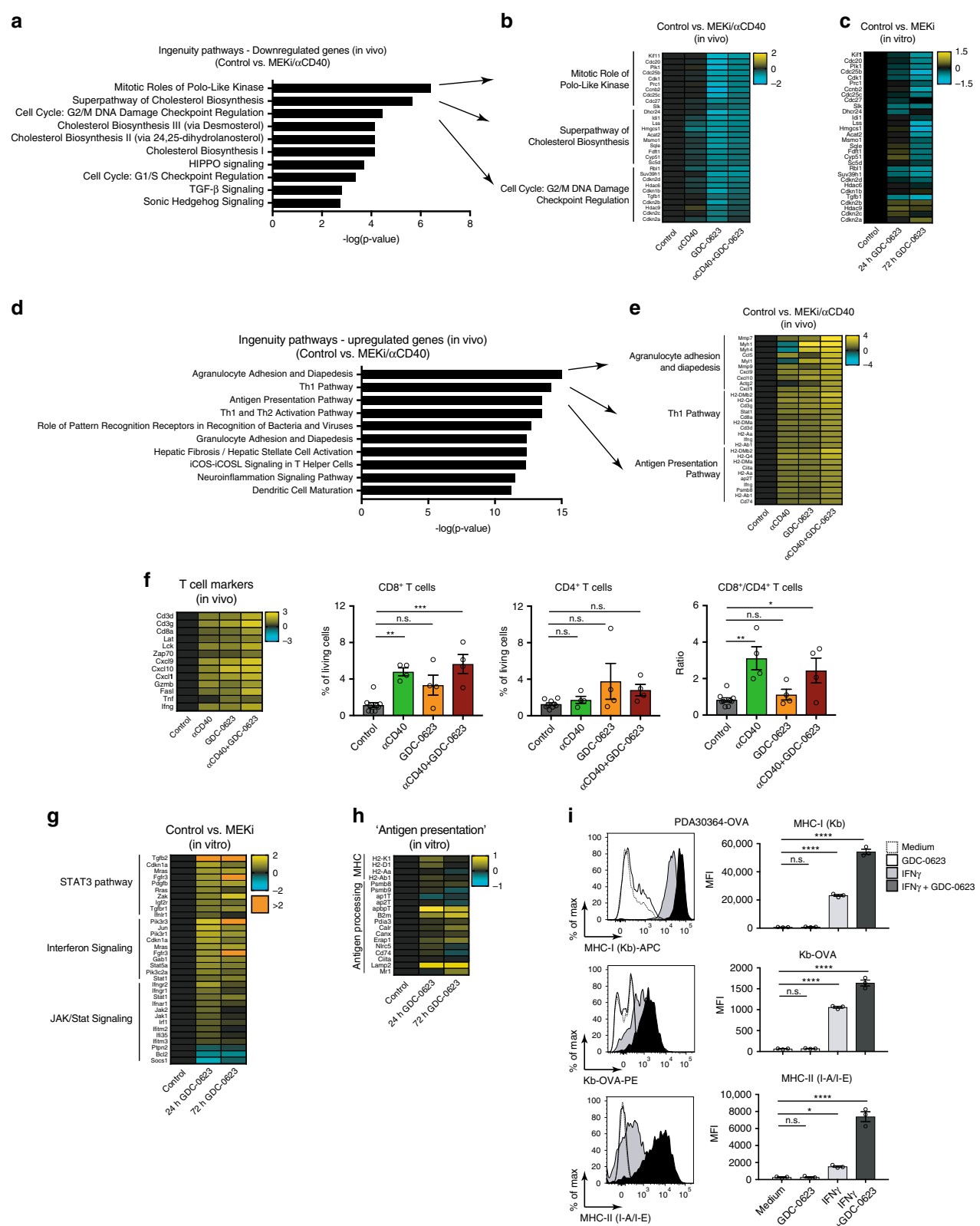

tumors treated with MEKi only. In conclusion, the in vivo antitumor impact of the single agent and combination treatments in the three different models, as well as the synergy between MEKi and anti-CD40 Ab action in this respect, are reflected by the cell cycle and biosynthesis associated gene signatures identified.

**Tumor transcriptome reflects pro-immunogenic action of MEKi.** In the same manner as above, we found that MEKi/CD40 Ab treatment of PDA30364 tumors activates multiple pro-inflammatory pathways involved in, amongst others, immune cell infiltration, the T cell immune response and antigen presentation (Fig. 6d–f). Interestingly, this immune signature is also induced in

**Fig. 6 Gene signatures reflect drug mechanism of action, including pro-immunogenic action of MEKi in PDA3034 tumors.** Whole-tumor transcriptome analysis of the impact of MEK inhibitor GDC-0623 and/or anti-CD40 Ab treatment on the T cell infiltrate in B16-OVA, MC-38, and PDA30364 tumors, as performed on four mice per treatment group of the experiments shown in Fig. 3a–c. **a–e** Genes signatures based on differentially regulated genes (two-component empirical Bayes test with $p < 0.05$ and Benjamini–Hochberg correction for multiple testing) between MEKi/CD40 Ab and untreated tumors followed subsequent IPA core analyses; log2 FC. **a** Top 10 canonical pathways based on $p$ value with focus on downregulated genes. **b** Top 10 differentially regulated genes of indicated pathways. **c** Gene expression changes of PDA30364 cell cultures treated with 100 nM GDC-0623 or vehicle for 24 and 72 hours with focus on genes identified in **b**. **d** Top 10 canonical pathways based on $p$ value with focus on upregulated genes. **e** Top 10 differentially regulated genes of indicated pathways. **f** T cell marker expression normalized to control group; log2 FC and flow cytometric analyses of tumor-infiltrating T cells isolated from PDA30364 tumors. Mean ± s.e.m., $n = 4$ mice per treatment group. One-way ANOVA with post hoc Dunnett's test (treatment groups vs. control). Significance levels are indicated by asterisks (*$p \leq 0.05$; **$p \leq 0.01$; ***$p \leq 0.001$; ****$p \leq 0.0001$). **g** Top 3 IPA 'cellular immune response' pathways after differential gene expression analyses (as above) of PDA30364 cell cultures treated with 100 nM GDC-0623 or vehicle for 24 and 72 hours. **h** Expression changes of genes associated with antigen presentation between aforementioned PDA30364 cell cultures; log2 FC. **I** Flow cytometric analysis of MHC levels on PDA30364-OVA tumor cells treated with 10 ng ml$^{-1}$ IFNγ and/or 100 nM GDC-0623. Upper panel MHC-I (H2-Kb), middle panel MHC-I-OVA (H2-Kb-OVA), lower panel MHC-II (I-A/I-E). Mean ± s.e.m, $n = 3$. One-way ANOVA with post hoc Dunnett's test (treatment groups vs. control).

PDA30364 tumors treated by either of the single drugs, supporting the notion that not only anti-CD40 Ab but also MEKi can have clear-cut pro-inflammatory impact. The biological relevance of this observation is illustrated by the striking correlation between the strength of the T cell signature and the CD8+ T cell count under the different treatment conditions (Fig. 6f).

In view of this finding, we evaluated the T cell signature in the context of CD8+ T cell count across the PDA30364, B16-OVA, and MC-38 models (Supplementary Fig. 10B). As expected, treatment with anti-CD40 Ab treatment induced this signature and increased CD8+ T cell count in all three models. In contrast, the impact of MEKi single-agent treatment ranged from activation in the PDA30364 model to suppression in the B16-OVA model, indicating that under certain conditions MEKi can suppress T cell immunity, as suggested by our in vitro T cell experiments (Supplementary Fig. 3 and Supplementary Fig. 4). This differential, tumor model-dependent impact of MEKi single treatment was also observed for the other pro-inflammatory pathways that were induced in treated PDA30364 tumors (Supplementary Fig. 10C) and is reminiscent of findings reported by Loi et al.[33] for the MMTV-neu model.

For the three tumor models examined in our study, the impact of MEKi on the immune gene signatures correlates with the single-agent cytostatic effect on these tumors, as well as with the in vitro IC50 values. In the highly sensitive PDA30364 tumor, strong suppression of the cell cycle and biosynthesis-related pathways (Fig. 6a–b) is associated with induction of pro-inflammatory genes (Fig. 6d–f). In the least sensitive B16-OVA tumor, marginal impact of MEKi single-agent treatment on the cell cycle and biosynthesis-related gene set (Supplementary Fig. 10A) is associated with marked suppression of the immune genes (Supplementary Fig. 10C). For the MC-38 tumor, these effects are intermediary. Taken together, these findings suggest that MEKi treatment is immunosuppressive unless accompanied by tumor cell death, an event known to be pro-immunogenic[12]. Importantly, in the less-sensitive B16-OVA and MC-38 tumors, MEKi-induced immunosuppression can be overcome by combining MEK inhibition with anti-CD40 Ab, resulting in synergistic antitumor efficacy (Fig. 3a–b), strong suppression of the cell cycle and biosynthesis-related pathways as well as induction of the pro-inflammatory gene sets (Supplementary Fig. 10).

Further evaluation of the transcriptome data of the PDA30364 in vitro cell cultures treated with MEKi also revealed upregulation of pro-inflammatory pathways, in particular pathways related to interferon signaling (Fig. 6g), suggesting that MEKi treatment increases the intrinsic immunogenicity of tumor cells. In view of a report by others showing that MEK inhibition can increase antigen presentation on tumor cells cells[29], we also checked for the upregulation of MHC and antigen-processing genes, but found

very little increase (Fig. 6h). Accordingly, in vitro treatment of PDA30364 cells with MEKi did not induce a significant upregulation of cell surface MHC expression. However, when MEKi treatment was combined with low levels of interferon-gamma (IFNγ), as are expected to be released by T cells in the TME upon stimulation, an induction of both MHC class I and II was observed that was significantly stronger than in cells treated with IFNγ only (Fig. 6i). The use of OVA-transduced variants of PDA30364 enabled us to show that the increase in MHC class I surface expression was mirrored by increased presentation of the H-2Kb-restricted OVA-derived SIINFEKL epitope, as detected the Kb/OVA-specific antibody 25-D1.16 (Fig. 6i). Importantly, these findings could be reproduced in the B16-OVA and MC-38 models (Supplementary Fig. 10D).

**Drug specific gene signatures as pharmacodynamic biomarkers.** The gene signatures discussed above were selected on basis of the impact of the MEKi/CD40 Ab combination treatment on PDA30364 tumors. In order to better discriminate between the in vivo impact of the single drugs, we identified pro-immune signatures that were induced by anti-CD40 Ab and MEKi single-agent treatment only. For anti-CD40 Ab, we composed a gene signature related to the activation of CD40-positive B cells, DCs, and other myeloid cells, which is dominated by the induction of genes encoding immunoglobulins and proteins involved in antigen processing and presentation (Fig. 7a left panel). The MEKi single-agent gene signature (Fig. 7a, left panel) features strong suppression of cell cycle genes and the induction of agranulocyte migration genes, mirroring tumor cell stasis and immune cell infiltration, respectively. The gene signature of the MEKi/CD40 Ab combination regimen, which is composed of genes for which synergistic induction by MEKi and anti-CD40 Ab can be seen, is dominated by the induction of genes associated with T cell activation, death receptor signaling and opsonization, reflecting the effector functions of CD8 + T cells and phagocytes, respectively (Fig. 7a, left panel).

Using these signatures, we evaluated the relationship between drug pharmacodynamics (PDs) and antitumor efficacy of the MEKi/CD40 Ab combination treatment and of a regimen in which anti-CD40 Ab was combined with GEM. The rationale for this is that prior work by others has shown synergistic antitumor action of anti-CD40 Ab and GEM in mouse models, including a transplantable PDA tumor setting[16,34]. However, in our PDA30364 model, we found no added value of combining anti-CD40 Ab and GEM, in spite of the fact that each of the single drugs had therapeutic impact (Fig. 7b). The difference between the efficacy of the MEKi/CD40 Ab and GEM/CD40 Ab modalities is illustrated in (Fig. 7c). Supplementation of albumin bound paclitaxel (nAbP) to this regimen, which was reported to enhance

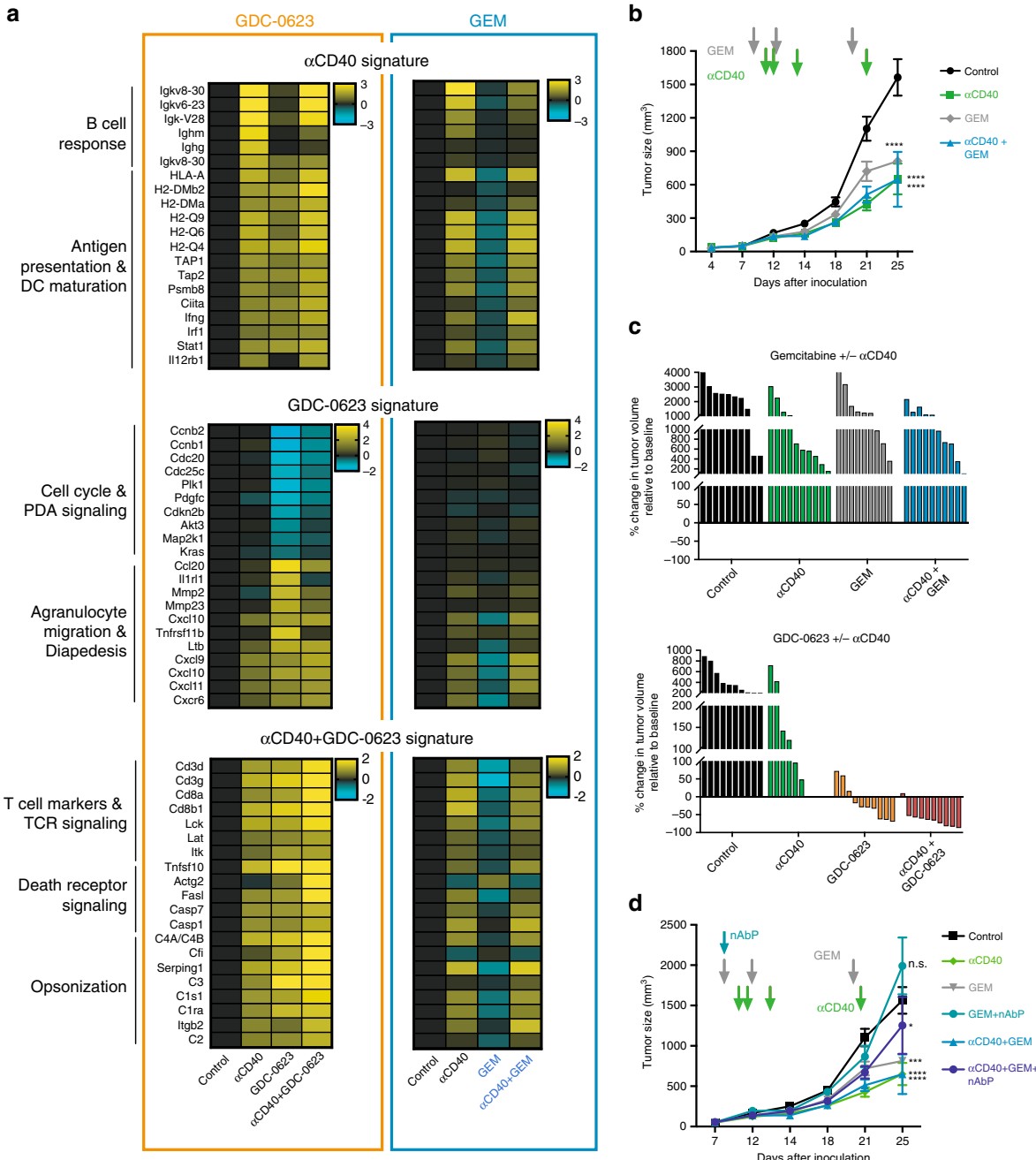

**Fig. 7 Gene signatures correlate with treatment efficacy and reveal drug mechanism of action. a** Gene signatures for each individual treatment regimen manually composed on the basis of differentially expressed genes between control and indicated treatment group of the MEKi/CD40 Ab experiment, as performed on four mice per treatment group of the experiments shown in Fig. 3a–c. In GEM/CD40 Ab PDA30364 experiments, 120 mg kg$^{-1}$ GEM was administered on treatment days 1, 5, and 12 and anti-CD40 Ab on days 3, 4, 6, and 14; mean ± s.e.m. Each treatment group consisted of at least seven animals. Four mice per treatment group were analyzed on day 22/23 after tumor inoculation. **b** Tumor growth of GEM/CD40 Ab or monotherapy groups. Mean±s.e.m. Two-way ANOVA with post hoc Fisher's LSD test (treatment groups vs. control; day 25). **c** Waterfall plots of GEM/CD40 Ab and MEKi/anti-CD40 Ab on day 22 after tumor inoculation. Each bar represents one individual mouse. Some data as in Fig. 3c and 7b. **d** Treatment of PDA30364 tumor-bearing mice with 120 mg kg$^{-1}$ GEM or nanoparticle albumin bound paclitaxel (nAbP). nAbP was administered on treatment day 1 and GEM was dosed on days 1, 5, and 12. Anti-CD40 Ab on days 3, 4, 6, and 14. Mean ± s.e.m. Four mice per treatment group were sacrificed on days 22/23 after tumor inoculation. Note, the control, anti-CD40 Ab, GEM, GEM/anti-CD40 Ab groups are the same as in **b**. Two-way ANOVA with post hoc Fisher's LSD test (treatment groups vs. control; day 25). Significance levels are indicated by asterisks (*$p \leq 0.05$; **$p \leq 0.01$; ***$p \leq 0.001$; ****$p \leq 0.0001$).

antitumor efficacy[14], also did not result in synergistic antitumor efficacy in our PDA30364 model (Fig. 7d). The difference in antitumor efficacy between the two treatments is closely reflected by differences in the gene signatures. In particular, the GEM and GEM/CD40 Ab regimens fail to induced a strong suppression of

the cell cycle and biosynthesis signature. Furthermore, whereas MEKi single-agent treatment induces multiple pro-immunogenic pathways, these are all markedly suppressed in GEM single-agent-treated tumors. Finally, the synergy in induction of T cell markers by the MEKi/CD40 Ab combination is not observed for the GEM/

CD40 treatment (Fig. 7a). Therefore, these transcriptome signatures provide meaningful PD biomarkers reflecting the difference in therapeutic efficacy of regimens.

## Discussion

In spite of the documented role of MAPK in TCR-downstream signaling[26,35–37] our data demonstrate that in vivo application of clinically relevant doses of MEK inhibitors, whereas effectively inducing regression of mutant KRAS-driven PDA30364 tumors, does not impair in vivo T cell priming, clonal expansion and effector function. Furthermore, MEKi has inhibitory effects on several immunosuppressive subsets in the TME, in particular CD4+ T-regulatory cells, M2-type macrophages, and MDSC. Importantly, these effects are not an indirect effect of MEKi-induced tumor cell death, because they are also observed during MEKi single-agent treatment of B16-OVA and MC-38 tumors where the impact on tumor growth is negligible. This direct nature of drug impact is further supported by in vitro experiments showing the greater sensitivity of MDSCs and M2-type macrophages, as compared with M1-type macrophages, to MEK inhibition. It is conceivable that these findings are related to the differential role of the MEK pathway in myeloid cell subsets. The action of the anti-CD40 Ab is highly complementary in this respect, in that it enhances antigen presentation by DCs, thereby indirectly stimulating CD8+ T cells, and promotes M1-like macrophages. Owing to this complementary drug action, the MEKi/CD40 Ab combination exerts synergistic antitumor efficacy as observed in three different syngeneic tumor models (Fig. 8).

Our initial experiments concerning the impact of MEK inhibitors on the T cell response were less encouraging, in that these in vitro assays showed strong suppression of T cell activation already at drug concentrations lower than those effective against tumor cells. Importantly, these assays involved co-cultivation of OVA-specific OT-I T cells with engineered APC expressing high levels of SIINFEKL/H-2Kb in combination with CD80. Similar

experiments by others have also shown that MEK inhibition abolishes T cell priming in conditions where T cell stimulation is limited to TCR and CD28 signaling[25,27,28,30,35], whereas this dependence on the MEK/ERK pathway can be overcome by costimulation through 4-1BB[29]. In our in vivo immunization assays, we combined delivery of the OVA antigen with systemic co-administration of agonist anti-CD40 Ab, which is known to induce DC maturation including the upregulation of a full array of co-stimulatory ligands[38,39]. This offers a plausible explanation for the disjuncture between our in vitro and in vivo T cell activation data, which has also been documented by Hu-Lieskovan and colleagues[25], and further supports the complementarity of MEK inhibitors and agonist anti-CD40 Ab in tumor therapy. Our data are in concordance with reports by others showing synergy in antitumor efficacy of MEK inhibitors with PD-L1 blocking antibodies and agonist antibodies targeting the T cell activatory receptors OX40 and 4-1BB[29,30,33].

Comprehensive analysis of the immune infiltrate in tumor samples in the context of treatment, as performed in our mouse models, presents a challenge in the clinical setting due to the need for fresh tissue samples. In view of this, we explored the use of whole-tumor transcriptome analysis in the three tumor models tested, which resulted in the identification of gene signatures that mirror the mechanism of action of the drugs, that reflect the differences in antitumor efficacy of the treatments in the three tumor models, and that correlate with the changes in immune infiltrate as observed by flow cytometry. For instance, the extent of suppression of the cell cycle-related pathways by MEKi treatment in the B16-OVA, MC-38 and PDA30364 tumors shows an excellent correlation with the low, moderate and strong impact of MEK inhibition on the growth of these tumors, respectively. Furthermore, the induction of pro-inflammatory gene signatures is only seen in conjunction with significant antitumor efficacy of the treatment regimen, in particular in all three tumors subjected to MEKi/CD40 Ab treatment as well as in PDA30364 tumors treated with MEKi only. Therefore, these signatures provide

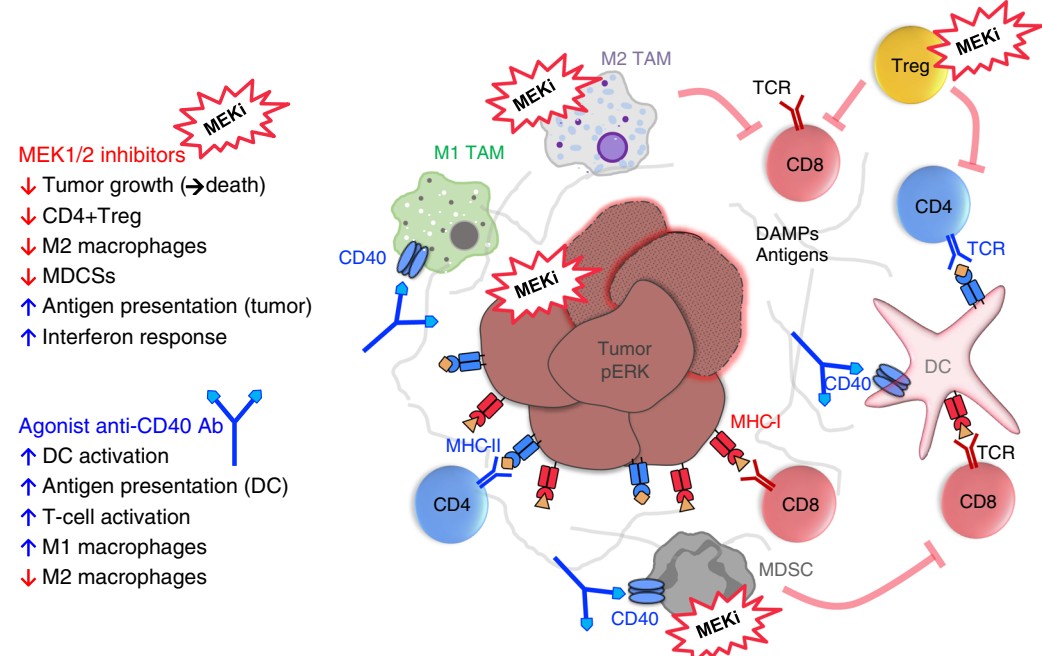

**Fig. 8 Graphical representation of MEKi/CD40 Ab antitumor mechanism.** MEK inhibitors induce tumor-intrinsic interferon response, enhance MHC-restricted antigen presentation and suppress M2 tumor-associated macrophages (M2 TAM), MDSCs and Tregs, but not M1 TAMs, DCs, and effector T cells. Agonist anti-CD40 provides co-stimulatory signal, increases antigen presentation, and stimulates CD4+ and CD8+ T cells. Complementary drug action of the MEKi/CD40 Ab combination exerts synergistic antitumor effects.

suitable surrogate endpoints for the evaluation of treatment efficacy, as well as PD biomarkers for monitoring drug exposure and impact in the tumor.

PD biomarker analysis by means of gene expression profiling provides a holistic view on the mechanism of action of drugs, beyond the commonly surveyed parameters. In this manner, we became aware of the stimulatory impact of MEKi on the interferon signaling pathway in in vitro cultured tumor cells, as well as the stimulatory effect of MEKi GDC-0623 on MHC class I and II expression at the surface of B16-OVA, MC-38 and PDA30364 tumor cells. These findings may be clinically relevant, because both HLA class I and II can also be induced to high levels on patient-derived PDA cell lines[40], whereas the T cell infiltrate in primary PDA tumors is relatively rich in CD4+ T cells[18]. Other human tumors, such as melanoma and glioma, can also be induced to express HLA class-II[41,42]. The implication of this finding is that both CD8+ and CD4+ T cells are involved in the T cell attack against the tumor, as we demonstrated for the PDA30364 model. This does not only reduce the risk of immune escape by downregulation/loss of MHC class I-restricted antigen presentation, as commonly found in various human and experimental tumors, including PDA[4,43], but also increases the repertoire of potential target antigens that can trigger T cell-mediated tumor recognition and destruction. Of interest in this respect is that a major fraction of the neo-epitopes encoded by the tumor mutanome may be presented in the context of MHC class-II[18]. Especially for cancer types with lower numbers of somatic mutations, such as PDA, co-expression of MHC class I and II may therefore result in an increase in tumor cell immunogenicity. Moreover, the induction of CD4+ T-helper responses against tumor antigens is essential for sustained CD8+ T cell responses, can neutralize pre-existing CD4+ T-regulatory cells as well as orchestrate activation and immune attack by innate immune cells[44–46].

In conclusion, the combination of MEK inhibitor GDC-0623 and agonist anti-CD40 Ab is a highly potent regimen for the treatment of tumors, especially for cancers such as PDA that are driven by mutated KRAS and heavily infiltrated by myeloid cells. By combining MEK inhibition with the CD40-downstream activation of immune effector mechanisms, limitations in the efficacy of MEK inhibition, alone or in combination with either GEM chemotherapy or receptor tyrosine kinase inhibition[22,23,47,48], can be overcome. Based on our experiments in the PDA30364 model, agonist anti-CD40 Abs are more effective than PD-1 blocking Abs in this respect, most likely because of their more profound impact on the immune cell infiltrate in the tumor. Furthermore, our data show that the combination of CD40 activation with targeted MEK inhibition is superior to the combination with GEM, which is typically used as a starting point for combination therapies in PDA. Although agonist CD40 Abs have been tested for their clinical efficacy as single agent and in combination with chemotherapy in several clinical indications, including pancreatic cancer, evidence for a therapeutic window is still lacking[10,11]. Our findings argue for exploring the therapeutic index of the MEKi/ CD40 Ab combination in mutant KRAS-driven tumors such as pancreatic cancer.

## Methods

**Informed consent and ethics committee approvals**. All animal procedures followed the institutional laboratory animal research guidelines and were approved by the governmental authorities (Regional Administrative Authority Karlsruhe, Germany).

**Patient ethics committee approval**. Primary human tumor tissue and blood samples were obtained from the European Pancreas Center of University Hospital Heidelberg. Informed written consent was obtained from the patients concerned before sample collection. The collection of patient samples was approved by the local ethics committee (Votum 301/2001, amendment 8 May 2012) and conducted in accordance with the declaration of Helsinki.

**Mice**. Mice were used in these studies as the least sentient species with an immune system. C57BL/6-Ly5.1 (CD45.1+, *Ptprca*) and NSG (NOD-*Prkdcscid*) mice were bred in animal facilities of the German Cancer Research Center. OT-I mice (C57BL/6-Ly5.2/CD45.2+; *Tg(TcraTcrb)1100Mjb*) were purchased from Charles River. For the experiments male mice were aged between 8 and 12 weeks. In immunization experiments, littermates of the same sex were randomly assigned to experimental groups at the start of the experiment. In tumor experiments, mice were grouped after tumor injection but prior to treatment in order to equalize tumor size within groups. Mice were held in individually ventilated cages in groups of up to five animals. Food and water were provided ad libitum. Mice were maintained on a 12 hour light/dark cycle and environmental enrichment was provided; temperature was maintained between 20 and 24 °C. Mice were visually checked daily if adverse effects were anticipated or if mice were nearing a endpoints.

**Cell lines and culture**. Human melanoma cell line A375[49], colon carcinoma cell line Colo205[50], and B-cell leukemia cell line JVM-2[51] were provided by professor Adelheid Cerwenka, Immunobiochemistry, UMM Mannheim, Germany. Human mammary carcinoma cell line MDA-MB-231[52] and hepatocellular carcinoma cell line U-87MG[53] were provided by professor Philipp Beckhove, Interventional Immunology, University Regensburg, Germany. Human pancreatic cancer cell line MIA PaCa-2[54] was provided by Dr. Nathalia Giese, European Pancreas Center, University Hospital Heidelberg, Germany. Human mammary carcinoma cell line MCF-7[55] was provided by professor Hellmut Augustin, Vascular Oncology and Metastasis, DKFZ, Heidelberg, Germany. Human mammary carcinoma cell line MDA-MB-468[56] was purchased from American Type Culture Collection (ATCC). Primary human patient-derived xenograft (PDX) cell lines (TIPC) were generated from human PDA tumor specimen by serial transplantation in NSG mice with subsequent in vitro culturing. Murine melanoma cell lines B16-F10[57] and B16-F10-OVA (B16-OVA) and murine colon carcinoma cell line MC-38[58] and MC-38-OVA were provided by Bayer Pharma AG Berlin, Germany. These cell lines were authenticated by whole-exome sequencing. The murine PDA cell line was generated from primary pancreatic tumors of PDA genetically engineered mouse model ElastTA/TetO-Cre *Kras*+/LSL-G12D *Tp53*+/LSL-R172H after cessation of doxycycline, which induces DNA recombination, and chronic pancreatitis induced by repetitive injections (three times per week, hourly injection for 6 hours) of 1 μg cholecystokinin analog cearulein (Sigma). This tumor induction protocol is based on the published work by Guerra et al.[59]. The presence of the KRAS and P53 driver mutations in the cell line was verified by whole-exome sequencing. PDA30364-OVA variant was generated via lentiviral transduction of PDA30364 with ecotropic platinum-e retroviral packaging cell line (Cell Biolabs, Inc.) using the pLenti6.3_3xFLAG-Ovalbumin-F2A-EGFP construct (Bayer Pharma AG). MEC.B7. SigOVA APCs were provided by Stephen Schoenberger, La Jolla Institute for Immunology, USA[60]. Human epithelial kidney cell line HEK293-F suspension cells were purchased from Thermo Fisher Scientific. A375, HepG2, JVM-2, MCF-7, MIA PaCa-2, U87MG, and PDA30364 were cultured in DMEM, 10% FBS, 100 units (U) ml−1 penicillin, and 100 μg ml−1 streptomycin (Thermo Fisher). PDA30364-OVA were cultured in DMEM, 10% FBS, 100 units (U) ml−1 penicillin, and 100 μg ml−1 streptomycin, 10 μg ml−1 blasticidin. Colo205, MDA-MB-231, and B16-F10 were cultured in RPMI, 10% FBS, 100 units (U) ml−1 penicillin, and 100 μg ml−1 streptomycin. MDA-MB-468 were cultured in Leibovitz's L15 medium, 10% FBS, 100 units (U) ml−1 penicillin, and 100 μg ml−1 streptomycin. MC-38 were cultured in RPMI, 10% FBS, 100 units (U) ml−1 penicillin, and 100 μg ml−1 streptomycin. B16-OVA cells were cultured in RPMI, 10% FBS, 100 units (U) ml−1 penicillin, and 100 μg ml−1 streptomycin, 2.5 μg ml−1 blasticidin. TIPC cell lines were cultured in advanced DMEM/F12 medium, 0.3% glucose, 12 μg ml−1 heparin, 5 mM HEPES, 2 mM L-glutamine, 100 units (U) ml−1 penicillin, and 100 μg ml−1 streptomycin. MEC.B7.SigOVA cells were cultured in DMEM, 10% FBS, 300 μg ml−1 G418, 300 μg ml−1 hygromycin. All adherenT cell lines were cultivated in either T25, T75, or T150 filter cell culture flasks at 37 °C and 5% CO2 in a humid incubator. Dependent on the growth rate, cells were split in a 1:20 to 1:50 ratio. HEK-293F cells were cultured in HEK293 Freestyle medium (Thermo Fisher) with a cell density 0.5–3 × 10^6 cells ml−1 at 37 °C, 8% CO2 orbital shaker platform rotating at 0.22 rcf. Cell lines are routinely tested for mycoplasma in house (McCT, Multiplexion). Prior to injection, cells were visualized under a microscope to ensure viability and verify that they exhibited the expected phenotype.

**Drugs and reagents**. GDC-0623 (order no. A-1181), GDC-0973-P1/S (cobimetinib; order no. A-1180), GDC-0973-P2/R, GSK1120212 (trametinib; order no. A-1258), MEK162 (binimetinib; order no. A-1128), GDC-0941 (pictilisib; order no. A-1017), AZD8186 (order no. A-1610), GSK2636711, MK2206 (order no. A-1909), BEZ235 (dactolisib; order no. A-1009) were purchased from Active Biochem. BAY 84-1236 (copanlisib) and BAY1082439 were provided by Bayer Pharma AG. Inhibitor stocks (10 and 100 mM) for in vitro use were generated by dissolving lyophilized inhibitor in dimethyl sulfoxide (DMSO). Inhibitors were stored at −20 °C until use and thawed only prior to use. For in vivo experiments, inhibitors

were suspended in methylcellulose (0.5% w/v)/tween (MCT) using a laboratory dispenser. Aliquots were prepared for several days and stored at 4 °C. GEM (Gemzar, Lilly Medical GmbH) and TEM (Temodal, Merck Sharp & Dhome) were provided by the University Clinic Pharmacy Heidelberg, Germany. Chemotherapeutic agents were freshly prepared in PBS and administered intraperitoneally. EndoGrade Ovalbumin protein was purchased from Hyglos (order no. 321001). Cytokines rm M-CSF (order no. 12343115), rm IFNγ (order no. 12343536) and rm IL-4 (order no. 12340042) were ordered from ImmunoTools. LPS (order no. L4391-1MG) was ordered from Sigma Aldrich. GM-CSF (order no. AF-315-03) and IL-6 (order no. AF-216-16) were purchased from PeproTech. Other reagents are detailed in the Key Resources Table.

**Antibody production and purification.** Chimeric agonist anti-mouse CD40 antibody (mIgG1, clone 3–23) sequences were obtained from professor Martin Glennie (University of Southampton, England). Codon optimization for the production in HEK (Homo sapiens) and CHO (Cricetulus griseus), as well as sub-cloning into pCEP4_A164 expression vector, was performed by GeneArt (Thermo Fisher Scientific). Plasmids stocks were maintained in form of regular MaxiPreps using the Endofree Plasmid Maxi kit (Qiagen) according to the manufacturer's protocol. HEK293-F suspension cells were transfected with the indicated constructs using the 293-Free transfection reagent kit (Novagen) with equimolar amounts of plasmid DNA for heavy and light chain. HEK293-F cells were seeded on day prior to transfection at $0.3 \times 10^6$ viable cells ml$^{-1}$ prewarmed FreeStyle293 expression medium. At the day of transfection, the cell number was adjusted to $1 \times 10^6$ cells ml$^{-1}$ in FreeStyle293 expression medium. Plasmid mix with equimolar amounts of heavy and light chain were diluted in mixed with 293-transfection reagent (Novagen). After 15 minutes incubation, the transfection mix was added to the cells in a dropwise fashion. Viability of cells was monitored for one week using a ViCell cell counter (Beckman Coulter). Transfected cells were cultured on a shaking platform at 37 °C, 8% $CO_2$ for ~1 week. Antibodies were purified with an ÄKTA pure chromatography system in conjunction with a HiTrap protein G column (1 or 5 ml) according to the manufacturer's (GE Healthcare) instructions. Bound antibodies were eluted from the columns with a glycine-based elution buffer (pH 2,7–3). Buffer exchange from Tris-HCl to sterile PBS was performed using a Slide-a-Lyzer dialysis cassettes (Thermo Fisher). Antibody concentration was determined using a NanoDrop 8000 device. Low levels of endotoxin (< 1 EU ml$^{-1}$) were confirmed by Limulus amebocyte lysate-based test with the Endosafe-PTS testing system (Charles River).

**Tumor cytotoxicity assays.** In all, 1000–10,000 tumor cells were seeded in 80 μl growth medium per well into a 96-well flat bottom plate using a VIAFLO96 handheld 96-channel electronic pipette device (Integra). The next day, the 10 mM inhibitor stock solutions were diluted to 500 μM in growth medium followed by a serial 1:10 dilution using the VIAFLO96 device. In all, 20 μl of diluted inhibitor were added to the cells. Medium and DMSO controls equivalent to the highest inhibitor concentrations were included. Cells were incubated for 72 h at 37 °C, 5% $CO_2$. At day 5 of the assay, cells and media were equilibrated to room temperature (RT) for 30 minutes. In all, 100 μl of CellTiter-Glo reagent (Promega) were added to each well and agitated on a shaking platform for 2 minutes. After 10 minutes of incubation, 100 μl were transferred to a 96-well white OptiPlate and luminescence was measured at 490 nm using the EnVision 2014 (Perkin Elmer) multilabel reader. Cell viability was calculated as percent of medium control plotted using GraphPad Prism 6.0. The concentrations of half-maximal inhibition ($IC_{50}$) were computed using a non-linear fit with bottom being constraint to greater than 0. The dose–response curve plots were exported and formatting was finalized using Adobe Illustrator imaging software.

**Isolation and CFSE labeling of OT-I T cells.** Spleens and lymph nodes (inguinal, axillary, and mesenteric) of OT-I mice (C57BL/6-Ly5.2/CD45.2+) mashed through a 100 μm cell strainer. Pelleted splenocytes were resuspended in ACK lysis buffer for red blood cell lysis. Splenocytes and lymph node cells were pooled and filtered through a 40 μm cell strainer. CD8+ OT-I T cell isolation was performed using a mouse CD8a+ T cell isolation kit (Miltenyi) according to manufacturer's instructions. Isolated CD8a+ OT-I T cells were labeled with 5 μM CFSE (Sigma). Successful CFSE staining was confirmed by flow cytometry.

**In vitro OT-I T cell assays.** MEC.B7.SigOVA APCs, which present SIINFEKL via H-2Kb and co-express co-stimulatory B7-1[60], were seeded into u-bottom plates ($2 \times 10^4$ cells in 100 μl T cell medium (Dulbecco's Modified Eagle's medium (DMEM), 10% FCS, 10 μM beta-mercaptoethanol, 0.5% HEPES, 100 units (U) ml$^{-1}$ penicillin, and 100 μg ml$^{-1}$ streptomycin) per well). After purification, 100 μl medium containing $2 \times 10^4$ CFSE-labeled OT-I T cells were added to MEC.B7.SigOVA. Inhibitors were added to the cultures. Final concentration 1 nM up to 10 μM, culture medium, as well as corresponding amounts of vehicle DMSO, were used as controls. The co-cultures were incubated for 72 hours at 37 °C and 5% $CO_2$. 72 hours after inhibitor addition, OT-I T cells were used for restimulation and the supernatant was used for IFNγ ELISA. For restimulation, OT-I T cells were resuspended in T cell medium containing 10 μg ml$^{-1}$ SIINFEKL peptide and a protein transport inhibitor (GolgiPlug; 1:1000). Cells were incubated for 6 h at 37 °C, 5% $CO_2$ and subsequently subjected to flow cytometric analyses. The data

were analyzed with the FlowJo software (version, FlowJo LLC). A representative gating strategy for T cells and myleloid cells is depicted in Supplementary Fig. 12. Resulting quantification of immune cell populations was visualized in bar graphs using GraphPad Prism 7 software. Histograms and pseudocolor plots were exported and formatting was finalized using Adobe Illustrator imaging software.

**In vivo OT-I T cell immunization assays.** These experiments were essentially performed as described by Glennie et al.[31,32]. One day prior to OT-I T cell infusion, C57BL/6-Ly5.1 mice were treated daily with small molecule inhibitors diluted in the corresponding vehicle solution for 6 days in total by oral gavage. On day 1, CFSE-labeled CD8a+ OT-I T cells (CD45.2+) were adoptively transferred via tail vein injection in C57BL/6-Ly5.1 mice. On day two, treated mice were immunized by a single intravenously injection of a mixture of full OVA protein (50 μg) and anti-CD40 antibody (50 μg) diluted in PBS. On day five, mice were killed 2 hours after the last inhibitor dose and OT-I downstream analyses were performed. Spleens were isolated mashed through a 70 μm cell strainer. After red blood cell lysis cells were again filtered through 40 μm strainers. For restimulation, OT-I T cells were resuspended in T cell medium containing 10 μg ml$^{-1}$ SIINFEKL peptide and a protein transport inhibitor (GolgiPlug; 1:1000). After an incubation period of 6 h at 37 °C and 5% $CO_2$, cells were subjected to flow cytometric analyses. The data were analyzed with the FlowJo software (version, FlowJo LLC). Resulting quantification of immune cell populations was visualized in bar graphs using GraphPad Prism 7 software. Histograms and pseudocolor plots were exported and formatting was finalized using Adobe Illustrator imaging software.

**In vivo OT-I T cell killing assays.** In all, 0.5 Mio OT-I T cells were transferred into B57BL/6-Ly5.1 mice via tail vein injection. In all, 24 h after transfer, recipient mice were immunized intravenously with 10 μg mIgG1 anti-CD40 antibody and 10 μg full OVA protein, dissolved in 150 μl endotoxin free PBS. Administration of small molecule inhibitors was performed by oral gavage between day 3 and 7. On day 8, splenocytes from C57BL/6 J (CD45.2+) donor mice were purified and divided in half for subsequent CFSE labeling. CFSE$^{high}$ control splenocytes were incubated with 5 μM CFSE CFSE$^{low}$ target T cells with 0.3 μM CFSE for 20 minutes at RT. One volume of FCS was added to the cell suspension and cells were centrifuged for 6 minutes at 350 g and 4 °C and washed with PBS. In all, 10 μg ml$^{-1}$ SIINFEKL was added to the CFSE$^{low}$ (target cells) and 10 μg ml$^{-1}$ P53 control peptide (AIYKKSQHM) to the CFSE$^{high}$ cells and cells were incubated at 37 °C for 30 minutes. Afterwards, cells were pooled and adjusted to $6 \times 10^6$ cells/150 μl PBS. In all, 150 μl of the cell suspension were injected into the tail vein of C57BL/6-Ly5.1 (CD45.1+) recipient mice. Splenocytes from recipient mice were isolated at day 8 and analyzed by flow cytometry. The data were analyzed with the FlowJo software (version, FlowJo LLC). Resulting quantification of immune cell populations was visualized in bar graphs using GraphPad Prism 7 software. Histograms and pseudocolor plots were exported and formatting was finalized using Adobe Illustrator imaging software.

**Tumor experiments and immunotherapy protocols.** Murine tumor cell lines were passaged at least four times after thawing and grown as a monolayer with a maximal confluency of 70–80%. Cells were detached using 0.25% trypsin-ethylenediaminetetraacetic acid (EDTA) solutions, centrifuged, and washed once with cell culture medium. $0.5 \times 10^5$ B16-OVA, $0.5 \times 10^6$ MC-38, $0.5 \times 10^6$ PDA30364 tumor cells were injected in PBS/Matrigel (1:1, Corning) subcutaneously into the flank of 6–12 weeks old C57BL/6-Ly5.1 mice. Tumor growth was measured twice a week with a caliper. The tumor volume was calculated by multiplying length by width by height. Tumor growth curves were generated using GraphPad Prism 7 software and formatting was finalized using Adobe Illustrator imaging software. Quantification of immune cell populations was visualized in bar graphs using GraphPad Prism 7 software. Treatment schedules and administration of drugs are indicated in the respective figure legends of each individual experiment. In general, treatment was initiated 1 weeks after tumor implant when tumors were palpable and started to grow. Mice were treated daily by oral gavage with small molecule inhibitors (200 μl per mouse) or intraperitoneally with chemotherapeutic drugs (200 μl per mouse). Control groups received 0.5% methylcellulose tween (MCT) or PBS, respectively. For combination therapy experiments, 200 μg of anti-CD40 mIgG1 (clone 3-23, in-house production) and 200 μg anti-PD-1 (BioXCell, clone RMP1-14, BP0146) were administered intraperitoneally. As isotype controls mouse IgG1 isotype control (BioXCell, clone MOPC-21, BP0083) and rat IgG2a (BioXCell, clone 2A3, BP0089) were used. Depletion of specific immune cell subsets were performed by administration of 200 μg depletory antibodies twice per week against CD8a (BioXCell, clone 2.43), CD4 (BioXCell, clone GK1.5) to deplete CD8 and CD4 T cells, respectively. Mice were killed if signs of distress were noticed, when termination criteria were reached, or analyses were performed at specific time points. For downstream analyses, such as flow cytometry and DNA/RNA analyses tissue pieces were processed individually. The flow cytometry data were analyzed with the FlowJo software (version, FlowJo LLC). Resulting quantification of immune cell populations was visualized in bar graphs using GraphPad Prism 7 software. Histograms and pseudocolor plots were exported and formatting was finalized using Adobe Illustrator imaging software.

**Tumor gene expression profiling**. RNA was isolated from tumor pieces of at least four animals per treatment group at the end of the treatment period and subjected to transcriptome analyses. Exact time points are indicated in the respective figure legends of each individual experiment. For this, 10–20 mg tumor material were transferred into a 2 ml Eppendorf tube containing RLT buffer supplemented with 1% β-mercaptoethanol and 2.8 mm steel beads (Precellys). Mechanical disruption was performed in a tissue homogenizer (TissueLyzer sample disrupter, Qiagen) at 30 Hz, 30 s for three times. Subsequent RNA extraction was performed using the RNeasy Mini kit (Qiagen) according to the manufacturer's instructions and with on-column DNase I digestion. RNA was eluted in RNase-free water (Ambion) and yields were determined using the NanoDrop 800 device. The integrity of the RNAs was assessed with an Agilent 2100 Bioanalyzer (Agilent Technologies GmbH, Berlin, Germany). Only samples with a ratio of absorption at 260 nm versus 280 nm in the range of 1.8–2.2, as well as an RNA integrity number above eight were considered for microarray analysis. Microarray gene expression analysis was carried out by the Microarray Unit of the Genomics and Proteomics Core Facility of the German Cancer Research Center (DKFZ) in Heidelberg, Germany. For the labeling procedure and associated 3′ in vitro transcription (3′ IVT) of total RNA for Affymetrix chips a 3′IVT Kit (GeneChip 3′IVT Plus Reagent Kit, Affymetrix) was used. In all, 200 ng of total RNA is turned into double stranded cDNA by reverse transcription using oligo-thymidine primers connected to the promoter sequence of the T7 bacteriophage. After second-strand cDNA synthesis, biotinylated copy RNA (cRNA) was generated by in vitro transcription using T7 RNA polymerase. 15 µg of each cRNA sample were fragmented and 11 µg were hybridized to the mouse 430 2.0 Gene Chip array (Affymetrix). After incubation for 16 hours using the 'Affymetrix GeneChip Hybridization, Wash, and Stain Kit', the chips were scanned in an Affymetrix GeneChip Scanner 3000. Raw data were exported as cell intensity files and analyzed using Chipster[61]. After quality control, samples were normalized by applying the robust multichip averaging (RMA) normalization method. Two groups of interest (indicated in the figure legends) were subjected to two-component empirical Bayes test with $p < 0.05$ and Benjamini–Hochberg correction for multiple testing. Resulting gene expression data were log2 transformed. Pathway analyses of differentially expressed genes were conducted with IPA (Qiagen Inc., https://www.qiagenbioinformatics.com/products/ingenuity-pathway-analysis[62]). For this, lists with differentially expressed gene were uploaded and subjected to expression core analysis. For heatmap representation of differentially expressed genes the group mean was calculated, log2 transformed and normalized to the control group. Heatmaps were generated using GraphPad Prism 7 software. The heatmaps were exported and formatting was finalized using Adobe Illustrator imaging software.

**Exome sequencing**. For exome sequencing and subsequent mutation calling murine tumor cell lines and corresponding controls were used (details in Supplementary Data 1). In case of the PDA30364 tumor cell line, the spleen of the original mouse was used as a germline control. For MC-38, the annotated mutation table was kindly provided by John Castle (Agenus) and for B16-OVA data from supplementary table 2 of Castle JC et al.[63], 'Exploiting the mutanome for tumor vaccines' was used for investigation of MAPK pathway-associated deriver mutations. Genomic DNA was extracted from tissues and pelleted cells using the DNeasy Blood & Tissue and the QIAamp DNA Mini kits (Qiagen, Hilden, Germany), respectively, quantified using the Qbit device (Thermo Fisher Scientific, Waltham, MA) and submitted for sequencing to the DKFZ Genomics Core Facility. There, exome libraries were generated using Agilent SureSelect kits (Agilent Technologies, Inc., Santa Clara, CA). For mouse samples, the SureSelectXT Mouse All Exon kit, for human samples the SureSelect Human All Exon v6 (without UTRs) was used. Sequencing was performed on an Illumina HiSeq 2500 machine (Illumina, San Diego, CA) with v4 chemistry in paired-end mode with 125 bp read length. For germline samples, we aimed for 40 million reads (one quarter Flow cell lane), whereas for tumor, xenograft, and tumor cell line samples we aimed for 75 million reads (0.5 lane). Resulting Fastq files were transferred to the HUSAR (Heidelberg Unix Sequence Analysis Resources) server and analyzed there. We used the Burrows-Wheeler aligner to map the Fastq files[64] to the human (hg19/GRCh37) or mouse (GRCm38) genomes, respectively. For patient-derived xenograft data, a hybrid human-murine reference genome was used for mapping. For read group addition and duplicate read marking we used Picard (http://picard.sourceforge.net/). We used GATK[65] for subsequent steps of data pre-processing according to the Best Practices workflow[66].

**Mutation calling**. Somatic mutations were called using MuTect2[67]. For detection of somatic gene deletions, we used a custom script utilizing BEDtools[68]. We annotated identified somatic mutations using ANNOVAR[69]. For the categorization of MAPK cascade-associated genes, we downloaded MAPK cascade gene lists (GO:0000165) for *Homo sapiens* and *Mus musculus* from the AmiGO 2 database[70] and matched them with genes carrying somatic non-synonymous mutations including stop codon gains/losses. A custom script for deletion detection ("deldec") is available in Supplementary Figure 11 and the reporting summary.

**Flow cytometry**. Tumor tissue (50–200 mg) was digested using a human tumor dissociation kit (Miltenyi) according to manufacturer's instructions in conjunction

with the gentleMACS Octo tissue dissociator (Miltenyi) with the program '37C_h_TDK_3'. After enzymatic digestion and homogenization, tumor cell suspensions were poured through a 100 µm pre-coated with 3% BSA/PBS. Spleens were isolated and mashed through a 100 µm cell strainer. Isolated splenocytes were resuspended in ACK lysis buffer (Lonza) in order to lyse red blood cells. Live-dead discrimination was performed with Zombie Aqua dead cell marker (Thermo Fisher). After an incubation period of 10 minutes at 4 °C, cells were washed twice in FACS buffer and resuspended 1:100 Fc receptor (FcR) triple block, consisting of α-CD16/32 clone 2.4G2 (BD Biosciences, cat. #553141), clone 93 (Biolegend, cat. #101302) and α-CD16.2 clone 9E9 (Biolegend, cat. #149502) diluted in fluorescence-activated cell sorting (FACS) buffer (PBS, 200 mM EDTA, 0.5% BSA). After 10 minutes blocking, extracellular staining was performed. After washing and centrifugation, pelleted cells were resuspended in antibody mixes and incubated at 4 °C for 25 minutes. Following antibodies against surface epitopes were used: CD45-PE/Dazzle594 (Biolegend, 1:1000, clone 30-F11, cat. #103145), CD3-FITC (Biolegend, 1:200, clone 17A2, cat. #100204), CD90.2-AF700 (Biolegend, 1:200, clone 20-H12, cat. #105320), CD8a-APC/Cy7 (Biolegend, 1:200, clone 53-6.7, cat. #100714), CD4-BV605 (Biolegend, 1:200, clone RM4-5, cat. #100548), CD25-BV711 (Biolegend, 1:200, clone PC61, cat. #102049), CD279 (Biolegend, 1:200, clone 29 F.1A12, cat. #135216), LAG3 (Thermo Fisher, 1:200, clone C9B7W, cat. #17-2231-82), TIM3 (Thermo Fisher, 1:200, clone RMT3-23, cat. #12-5870-82), CD11b-FITC (Biolegend, 1:1000, clone M1/70, cat. #101206), F4/80-BV605 (Biolegend, 1:200, clone BM8, cat.#123133), Gr1-PE/Dazzle594 (Biolegend, 1:1000, clone RB6-8C5, cat. #108452), Ly6G-AF700 (Biolegend, 1:1000, clone 1A8, cat. #127622), Ly6C-FITC (Biolegend, 1:1000, clone HK1.4, cat. #128005), CD40-PE (Biolegend, 1:200, clone 3/23, cat. #124610), I-A/I-E-APC/Cy7 (Biolegend, 1:1000, clone M5/114.15.2, cat. #107627), CD86-PE/Cy7 (Biolegend, 1:1000, clone GL-1, cat. #105014), CD80-BV605 (Biolegend, 1:1000, clone 16-10A1, cat. #104729), H-2Kb-APC (Biolegend, 1:1000, clone AF6-88.5, cat. #116518), H2-Kb/SIINFEKL-PE (Biolegend, 1:1000, clone 25-D1.16, cat. #141603). In case of staining of intracellular antigens, cells were fixed using the Transcription Factor Buffer set (BD) according to the manufacturer's instruction. Intracellular antibodies were diluted in Perm-Wash buffer. Following antibodies were used to detect intracellular epitopes: Foxp3-eFl450 (Thermo Fisher, 1:100, clone FJK-16s, cat. #48-5773-82), IFNγ-BV421 (Becton Dickinson, 1:1000, clone XMG1.2, cat. #563376), TNFα-PE (Biolegend, 1:1000, clone MP6-XT22, cat. #506306), CD206-BV421 (Biolegend, 1:200, clone C068C2, cat. #141717), iNOS-APC (Thermo Fisher, 1:200, clone CXNFT, cat. #17-5920-82), Ki67-APC (1:200, clone 16A8, Biolegend, cat. #652406). In order to monitor the effector cytokine production of TILs, single cells suspensions were generated as described above and incubated in T cell medium containing 1:1000 dilution of GolgiPlug for 5 hours at 37 °C supplemented with 100 ng ml$^{-1}$ PMA 500 ng ml$^{-1}$ Ionomycin. Cells were subsequently stained for T cell markers and intracellular effector cytokines IFNγ and TNFα. For cell cycle analyses, cells were treated with small molecule inhibitors or DMSO and incubated at 37 °C, 5% CO$_2$ for 72 h, detached by addition 0.25% trypsin-EDTA and resuspended in 200 µl ice cold (pre-chilled at −20 °C) 70% ethanol for fixation for 2 hours at −20 °C. After fixation, cells were rinsed with FACS buffer and stained with Ki67 antibody for 30 minutes at RT. After this step, cells were washed with FACS buffer and resuspended in PI staining solution (50 µl PI + 12.5 µl RNAse+16 µl MgCl$_2$ in PBS). Twenty minutes after incubation at RT samples were recorded with a BD LRS Fortessa flow cytometer. The flow cytometry data were analyzed with the FlowJo software (version, FlowJo LLC). Resulting quantification of immune cell populations was visualized in bar graphs using GraphPad Prism 7 software. Histograms and pseudocolor plots were exported and formatting was finalized using Adobe Illustrator imaging software.

**Ex vivo myeloid cell assays**. Macrophage precursors were isolated from bone marrow of C57BL/6-Ly5.1 mice and differentiated ex vivo with cytokines into M1- or M2-like macrophages, respectively. For this, bones from femurs and tibias were isolated and crushed with a mortar in RPMI-1640 medium. The medium containing bone marrow cells were filtered through a 70 µm cell strainer. For red blood cell lysis were resuspended in ACK lysis buffer (Lonza) for 2 minutes at RT. Cells were resuspended in macrophage medium (RPMI-1640 supplemented with 10% FCS, 100 U ml$^{-1}$ penicillin, 100 µg ml$^{-1}$ streptomycin and 20 ng ml$^{-1}$ M-CSF). On day 5, non-adheren T cells were spun down and medium was replaced by fresh macrophage medium containing M-CSF. On day 7, macrophages were polarized into M1- or M2-like macrophages by replacing the basal macrophage medium by M1- (RPMI-1640 with 10% FCS, 100 U ml$^{-1}$ penicillin, 100 µg ml$^{-1}$ streptomycin and 20 ng ml$^{-1}$ M-CSF supplemented with 1 ng ml$^{-1}$ LPS and 1 ng ml$^{-1}$ IFNγ) and M2-polarization medium (RPMI-1640 with 10% FCS, 100 U ml$^{-1}$ penicillin, 100 µg ml$^{-1}$ streptomycin and 20 ng ml$^{-1}$ M-CSF supplemented with 2.5 ng ml$^{-1}$ IL-4), respectively. The suppressive impact of polarized macrophages was evaluated in co-culture experiments with pre-activated OT-I T cells. Macrophages were seeded into a 96-well flat bottom at different T cell/macrophage ratios. After polarization of macrophages, MEC.B7.SigOVA 24 h pre-activated OT-I T cells were added to the culture and incubated 48 hours at 37 °C. OT-I T cells were restimulated with SIINFEKL peptide for 5 hours in the presence of GolgiPlug. MDSC precursors were isolated from bone marrow of C57BL/6-Ly5.1 mice and differentiated ex vivo with cytokines into MDSCs. For this, bones from femurs and tibias were isolated and bone marrow was washed out with PBS by using a 27 G

needle. The PBS containing bone marrow cells was filtered through a 100 μm cell strainer. Cells were resuspended in ACK lysis buffer (Lonza) in order to lyse red blood cells. Cells were resuspended in MDSC medium (RPMI-1640 GlutaMax supplemented with 10% FCS, 100 U ml$^{-1}$ penicillin, 100 μg ml$^{-1}$ streptomycin, 10 mM HEPES, 50 μM β-Mercaptoethanol, 1 mM NEAA, 1 mM sodiumpyruvate, 40 ng ml$^{-1}$ IL-6, and 40 ng ml$^{-1}$ GM-CSF).

**Mouse IFNγ ELISA**. Supernatant IFNγ levels of MEK inhibitor treated OT-I T cell cultures were analyzed using BDOptEIA Kit (BD) according to manufacturer's instructions. In brief, plates were coated with the capture antibody (anti-mouse IFNγ monoclonal antibody, 1:250) at 4 °C overnight. The next day, the supernatant was discarded and the plates were washed five times with washing buffer. Assay diluent was used for block of unspecific binding. After an incubation period of 1 hour at RT, the assay diluent was discarded and the plates were washed five times with washing buffer. For absolute quantification, IFNγ standard was included. The plates were incubated for 2 hours at RT in the dark and washed five times with washing buffer. The detection antibody (biotinylated anti-mouse IFNγ monoclonal antibody) and the enzyme reagent (streptavidin-horseradish peroxidase conjugate) were diluted 1:250 in assay diluent and added to the wells. After incubating the plates for 1 hour at RT in the dark, the detection-reagent mix was discarded, the plates were washed five times with washing buffer and substrate solution (hydrogen peroxide and 3,3′, 5,5′ tetramethylbenzidine) was added. The plates were incubated for ~10 minutes. The reaction was stopped by adding sulfuric acid solution (2 N) and the absorbance measured at 450 nm with the plate reader (PerkinElmer). The IFNγ concentration of each sample was calculated in GraphPad Prism software by generating a hyperbola standard curve. Bar graphs were exported and formatting was finalized using Adobe Illustrator imaging software.

**Peptide synthesis**. The OVA (257-264) SIINFEKL and P53 control peptides AIYKKSQHM were chemically synthesized and purified at the German Cancer Research Center by professor Dr. Stefan Eichmueller. For the solid phase synthesis the Fmoc-strategy in a fully automated multiple synthesizer Syro II (MultiSyn Tech, Germany) was employed. The synthesis was carried out on preloaded Wang-Resins. As coupling agent 2-(1H-Benzotriazole-1-yl)-1,1,3,3- tetramethyluronium hexafluorophosphate was used. The material was purified by preparative high-performance liquid chromatography (HPLC) on a Kromasil 100–10 C 10 μm 120 A reverse phase column (20 × 150 mm) using an eluent of 0.1% trifluoroacetic acid in water and 80 % acetonitrile in water. The peptide was eluted with a successive linear gradient of 25% B to 80% B in 30 minutes at a flow rate of 10 ml min$^{-1}$. The fractions corresponding to the purified protein were lyophilized. The purified material was characterized with analytical HPLC and mass spectrometry (Thermo Finnigan LCQ).

**In vitro MEKi immunogenicity assays**. Murine tumor cell lines were treated with 100 nM MEKi GDC-0623 at IC50 for 24 and 72 hours. For this, cell numbers and volumes were upscaled from 96-well format to T75 format (450,000 cells). GDC-0623 was added five times concentrated to reach a final concentration of 100 nM. DMSO at the same concentration served as control. RNA was isolated as described above and subjected to mouse 430 2.0 Gene Chip array (Affymetrix) analyses. For MHC class I and class-II surface expression and OVA presentation analyses, murine tumor cells were treated 100 nM GDC-0623 and 10 ng μl$^{-1}$ IFNγ for 48 hours and analyzed by flow cytometry. The flow cytometry data were analyzed with the FlowJo software (version, FlowJo LLC). Resulting quantification of immune cell populations was visualized in bar graphs using GraphPad Prism 7 software. Histograms and pseudocolor plots were exported and formatting was finalized using Adobe Illustrator imaging software.

**Western blot analysis**. Murine tumor cell lines were treated with 500 nM MEKi GDC-0623 or DMSO as control for 1 hour. The medium was removed, cell once washed with cold PBS and lysed in radioimmunoprecipitation buffer (Santa Cruz, sc-24948) containing phenylmethanesulfonyl fluoride, sodium orthovanadate, and protease inhibitor cocktail according to manufacturer's instructions. The samples were incubated 20 min on an orbital shaker for 20 min. After centrifugation (13,000 × g for 30 min at 4 °C) the cell lysate containing supernatant was used for sodium dodecyl sulphate-polyacrylamide gel electrophoresis (SDS-PAGE). Protein amounts were determined by BCA test (Pierce, #23225) according to manufacturer's instructions. Lysates were mixed with Laemmli buffer (Bio-Rad, #1610727) containing 10% beta-mercaptoethanol, heated up to 95 °C for 5 min and loaded onto Mini-Protean TGX stain-free 4–20% protein gel (Bio-Rad, 456-8095). The SDS-PAGE was run for 10 min at 80 V followed by 60 min at 100 V. The transfer was performed using a Trans-Blot Turbo Transfer System (Bio-Rad, #1704150) in combination with Trans-blot Turbo Mini polyvinylidene difluoride (PVDF) Transfer membranes (Bio-Rad, #1704156) according to manufacturer's instructions. The PVDF membrane was rehydrated for 20 s in 100% MeOH and used for antibody staining. Membranes were washed in TBST (TBS with 0.1% Tween-20) and blocked with 5% BSA in TBST. Staining with primary antibodies was performed overnight at 4 °C and with secondary antibodies 2 h at RT. Following primary and secondary antibodies and dilutions were used: anti-mouse pERK1/2 (Cell signaling, clone 20G11, cat. #4376, 1:1000), anti-mouse ERK (Cell

signaling, clone 137F5, cat. #4695, 1:1000), anti-mouse Gapdh (GeneTex, cat. #GTX100118, 1:1000), Goat anti-rabbit IgG-HRP (Cell signaling, cat. #7074, 1:3000). Clarity Western ECL substrate (Bio-Rad, cat. #170-5060) was used for chemiluminescence reaction according to manufacturer's instructions. Images were processed (cropped and adjusted for contrast and brightness) using the ImageLab Software (Version 6.0.1, Bio-Rad). Processed images were exported and formatting was finalized using Adobe Illustrator imaging software.

**Statistical analysis**. Data analysis throughout was performed using GraphPad Prism 7.0 for Mac software. Where averages and error bards are indicated these are means and standard error of the mean or standard deviation where specified. Individual data points for in vivo experiments refer to one individual mouse. Statistical analyses for multiple comparisons are by one-way or two-way ANOVA with Tukey's, Dunnett's or Fisher's LSD post hoc multiple comparison test as indicated in the figure legends. $p < 0.05$ is considered significant; *$p < 0.05$, **$p < 0.01$, ***$p < 0.001$, ****$p < 0.0001$. Numbers ($n$) are defined in the relevant figure legends. Group sizes for murine experiments were determined by previous experience with similar models and on the basis of power analyses performed in collaboration with biostatistics department of German Cancer Research Center. For microarray whole-transcriptome analyses two groups of interest, as indicated in the figure legends, were subjected to two-component empirical Bayes test with $p < 0.05$ and Benjamini–Hochberg correction for multiple testing.

**Reporting summary**. Further information on research design is available in the Nature Research Reporting Summary linked to this article.

## Data availability
The mouse tumor and in vitro trancriptome data have been deposited in the GEO database under the accession codes GSE144128 (PDA30364: GEM/CD40 tumor), GSE144139 (MC-38: MEKi/CD40 tumor), GSE144145 (PDA30364: MEKi/CD40 tumor), GSE144146 (B16-OVA: MEKi in vitro), GSE144161 (MC-38: MEKi in vitro), GSE144166 (PDA30364: MEKi in vitro), GSE144570 (B16-OVA: MEKi/CD40 tumor) [https://www.ncbi.nlm.nih.gov/geo/]. The mouse exome data have been deposited in the ENA database under the accession number ERP119708 [https://www.ebi.ac.uk/ena]. The human exome data have been deposited in the EGA database under the accession code EGAS00001004196 [https://www.ebi.ac.uk/ega/home]. All the other data supporting the findings of this study are available within the article and its supplementary information files and from the corresponding author upon reasonable request. A reporting summary for this article is available as a Supplementary Information file.

## Code availability
The custom deletion detection ("deldec") script for exome analysis is depicted in Supplementary Figure 11 and available in the reporting summary. The chipster code for microarray-based transcriptome analysis is available in the reporting summary as well.

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

## Acknowledgements

We acknowledge the support of the microarray unit of the DKFZ Genomics and Proteomics Core Facility, the Center for Preclinical Research, DKFZ, the DKFZ–Bayer Alliance, and the Small Animal Imaging Facility, DKFZ. We thank the DKFZ-Heidelberg Center for Personalized Oncology (DKFZ-HIPO) for technical support and funding through HIPO_K28E. Further, we would like to thank Martin Glennie, Claude H.T. Chan and their team for providing the anti-mouse CD40 Ab 3/23 muIgG1 and for critically reviewing the manuscript, Oezlem Tureci and Anreas Fregin for help in optimizing the production of agonist Ab, Kurt Reifenberg and his staff for excellent support with in vivo experiments, Manfred Jugold and his team for the imaging of PDA GEMM tumors, Carmen Guerra and Mariano Barbacid for provision of the PDA GEMM model, Stephen Schoenberger for providing the MEC.B7.SigOVA cell line, Dariya Jäger and Oliver Strobel for providing GEM, TEM, and nAbP, Oliver Politz and Bayer team for providing BAY 84-1236, Isabel Poschke and her team for providing the PDX tumor cell lines, Stephen Gould for his technical advice concerning formulation of small molecule inhibitors, John Castle, Lars Roese for help with authentication of the murine tumor cell lines and Michael Volkmar for further assistance with analysis of the expression microarray data.

## Funding

This work was financially supported by the K.H. Bauer foundation (to R.O., C.V., I.N., D.B., C.V., and M.V.), the Graduate school of the German Cancer Research Center (to D.B. and J.M.), The European Union (FP7 program, project # 602262 'Immunostimulatory Agonist antibodies for Cancer Treatment' to R.O.), the Helmholtz Foundation (Immunology & Inflammation Future Theme to T.H. and M.V.) and the Federal Ministry of Education and Research (Junior systems biology research consortium 'Dynamics of the tumor-infiltrating lymphocyte repertoire in melanoma and pancreatic cancer' to O.S. and S.W.).

## Author contributions

R.O. and D.B. designed the study and wrote the manuscript. D.B., T.H., and S.B. designed and performed experiments, and analyzed data. D.B. performed microarray arrays and analyzed the data. D.B. performed statistical analyses. J.D., C.S., S.B., T.H., S.B., S.L.B., and J.H.N. performed in vitro tumor cytotoxicity and OT-I T cell assays with small molecule inhibitors and analyzed the data. J.D. and T.H. performed myeloid cell assays. J.H.N. performed in vitro antigen presentation assays and analyzed the data. D.B., C.V., and T.H. performed OT-I T cell in vivo assays and analyzed the data. C.S. performed the qPCR experiments. D.B. and J.D. performed tumor cell cycle analyses and western Blots. D.B. generated the PDA30364 cell line. D.B. and S.W. generated the PDA30364-OVA cell line. D.B., A.S.D., T.H., J.M., C.V. produced the anti-CD40 antibody. D.B., T.H., J.M., S.B., J.H.N., C.V., J.D., I.N., S.W., S.B, A.S.D. performed in vivo therapy experiments and subsequent flow cytometric analyses. M.V. and O.S. interpreted data and gave scientific input.

## Competing interests

The authors declare no competing interests.
