## [Peer Review File · Nature Communications]

Reviewers' comments:

Reviewer #1 (Remarks to the Author):

This is a preclinical study by Baumann et al demonstrating synergy between CD40 agonist antibodies and small molecule MEK inhibitors. Overall the data tell a complete story - however while the CD40 angle is novel, most of the effects of MEK activation have been published by Loi et al CCR 2016 and Ebert et al, Immunity 2016. the authors should address the two major points below to enhance rigor and novelty.

Major:

1) A fair amount of the data should be supplementary:

First, Figure 1 is not very exciting since the growth curves for the cell lines really don't play much into the central thesis of the paper.

Many of the transcriptional effects of MEK inhibition are duplicative of those already identified in Loi et al, CCR 2016, and not cited. These data are not really novel except that they are in additional tumor models than those described. Can the authors take the next step and show that any of these features are critical to the activity of MEKi on T cell activity? This would be an advance.

2) in Figure 3, PD-1 or L1 should be shown as a comparator - the immunogenic effects of MEK inhibitors are now established, in combination with PD-1/L1, so - is this combination better or just an additional possible option - either is fine, but the relative levels of activity should be described

Minor:

1) There are some inaccuracies as to the state of the field in the introduction. The authors cite the limited utility of immune checkpoint blockade in the opening statement of the introduction. The statement is a bit misleading. ICB has been approved in a vast variety of tumor types, not just lung and melanoma. This is a strange tennet since the authors do not test ICB. Later, "while immune checkpoints appear primarily suitable for mobilizing existing T cell responses..." - this statement reflects what is known about PD-1/L1, but not CTLA-4.

2) The drug name GDC-0623 is erroneously reported in Figure 1 legend. (0632)

3) Figure 2A is missing a control - what about OVA without CD40?

4) Details on figure 2C are missing - what are the target cells? (also spelled typo - targeT-cell on page 9

Reviewer #2 (Remarks to the Author):

In this study, Baumann et al reported a synergy between MEK inhibitors and agonist anti-CD40 immunostimulatory antibodies. Based on literature and multiple in vitro cancer cell line models, the authors selected a MEKi GDC-0623 as an optimal cytostatic small molecule inhibitor for immune-oncology applications. While GDC-0623 inhibited T-cell proliferation and cytokine production in vitro, at tumor-effective dosages GDC-0623 did not significantly impair DC and T-cell functions in the in vivo OT-I T-cell immunization assays nor in the in vivo OT-I T-cell killing assays. GDC-0623 and agonist anti-CD40 Ab showed synergistic anti-tumor efficacy in three syngeneic

tumor models, B16-OVA (melanoma), MC-38 (colorectal cancer), and PDA30364 (pancreatic cancer). The authors showed that in the PDA30364 model, the synergy was T-cell dependent. In addition, GDC-0623 was more toxic for the immunosuppressive M2 macrophages and MDSCs than the pro-inflammatory M1 macrophages. To understand the mechanism of the synergy, the authors performed RNAseq analyses for GDC-0623/agonist anti-CD40 Ab-treated tumor samples. Based on differential mRNA expression levels, cell cycle and biosynthesis-related pathways were down-regulated by GDC-0623, and T cell markers were up-regulated by agonist anti-CD40 Ab in all three syngeneic models. Furthermore, using the pancreatic cancer PDA30364 syngeneic tumor model, the authors also developed gene signatures for GDC-0623, agonist anti-CD40 Ab, and their combination, that correlated with treatment efficacy. These findings are significant and important for developing effective therapeutic strategies for MAPK pathway-dependent cancers. The manuscript is well written. Considering the following suggestions may help strengthen the proposed model and the manuscript.

Major comments:

1. The author indicated that the synergy of MEKi and agonist anti-CD40 Ab in tumor therapy was "T-cell mediated". However, among the three syngeneic tumor models used in this study, only in the pancreatic cancer PDA30364 model MEKi enhanced T cell proliferation and activation as a single agent, and showed synergy with anti-CD40 Ab in T cell activation. In the MC-38 model, MEKi did not show significant effect on T cell proliferation nor synergy with anti-CD40 Ab on T cell proliferation (supplementary Fig. 5B). In the B16-OVA model, MEKi clearly suppressed T cell proliferation and showed antagonism with agonist anti-CD40 Ab in T cell proliferation (supplementary Fig. 5B). In addition, MEKi decreased the expression of many genes involved in immune cell infiltration, T-cell immune response and antigen presentation in both B16-OVA and MC-38 tumors (supplementary Fig.5C). Therefore, mechanisms other than T cell proliferation and activation may be involved in the synergy of MEKi and agonist anti-CD40 Ab in the MC-38 and B16-OVA models. Based on these results, I would recommend to remove "T-cell mediated" from the manuscript title.
2. At the beginning of the Results section, the authors determined GDC-0623 as the optimal cytostatic compound for immune-oncology application. However, the assay (CellTiter-Glo assay) they used to test the compounds was an ATP-based assay, which could not distinguish between cytostasis and cytotoxicity. In fact, GDC-0623 was cytotoxic for M2 macrophages, as indicated by the substantial increase in the sub-G1 population in Fig. 5C. The author should perform a viability assay, such as trypan-blue staining or PI-cell cycle analysis, to confirm the cytostatic feature of GDC-0623 in their cancer cell models.
3. In Fig. 1c, the authors tested the sensitivity of primary patient-derived PDA cell lines to MEKi GDC-0623 to support the point that this inhibitor has superior efficacy in KRAS-driven tumors. For that, the authors should provide the KRAS mutation status of these patient-derived PDAC cell lines.
4. In all data in Fig.2, negative controls without OVA and anti-CD40 Ab injection should be included.
5. Data in Fig. 3E showed that either CD4+ T cells or CD8+ T cells were sufficient for the synergy of MEKi and anti-CD40 Ab in the PDA30364 model. Depletion of either population did not significantly affect the synergy. This observation is further supported by the effect of anti-CD40 on activating both CD4+ T cells and CD8+ T cells (Fig. 4E). This is a very interesting observation that should be further discussed.
6. The same CD8+ T cell data were shown twice in the manuscript (Fig. 4C and Fig. 6F/supplementary Fig. 5B). I recommend to remove the CD8+ T cell data from Fig. 4C and keep them in Fig.6F and supplementary Fig. 5B.
7. In Fig.6I and supplementary Fig. 5D, the authors showed that, in cell culture models, MEKi GDC-0623 and IFN γ synergistically up-regulated MHC expression at the cell surface. This could be an important mechanism of the synergy of MEKi and anti-CD40 Ab in in vivo syngeneic tumor models. Isogenic IFN γ receptor deletion tumor models could be used to determine the role of IFN γ in mediating the synergy in vivo.
8. In the proposed model in Fig.8, the authors indicated KRASG12D tumors. However, among the

three syngeneic tumor models used in the study, only PDA30364 carries KRASG12D. The authors should revise the model.

Minor comments:

1. In the histogram plots in Fig.2 and supplementary Fig. 2, the author should indicate what different color curves are for.
2. Dosages and concentrations of agents used in experiments should be included in the figure legends.
3. Significance needs to be added to Fig. 6I and supplementary Fig. 5D.

Point by point response to Reviewers' comments:

Reviewer 1 (Remarks to the Author):

This is a preclinical study by Baumann et al demonstrating synergy between CD40 agonist antibodies and small molecule MEK inhibitors. Overall the data tell a complete story - however while the CD40 angle is novel, most of the effects of MEK activation have been published by Loi et al CCR 2016 and Ebert et al, Immunity 2016. the authors should address the two major points below to enhance rigor and novelty.

Major:

1) A fair amount of the data should be supplementary:

1A) First, Figure 1 is not very exciting since the growth curves for the cell lines really don't play much into the central thesis of the paper.

RESPONSE:

We have condensed the in vitro drug action data to the curves for GDC-0632 and the small summary table (revised Fig. 1A-B), and have pasted the other data into Suppl. Fig 1. We decided to keep at least part of the data in the core figures in view of comments #2 and #8 by Reviewer 2 concerning more detailed specification of the three tumor models used. In response to the latter comments, we added the following data:

- Revised Fig. 1C (NEW DATA), showing that all three tumors express elevated pERK levels that are profoundly inhibited by MEKi
- Revised Fig. 1D-E, showing the impact of MEK inhibition with respect to cytotaxis and cytotoxicity for the three tumor cell lines concerned (formerly Suppl. Fig. 3B-C)

Notably, due to this and further changes in the manuscript the numbering of the supplemental figures has changed considerably.

1B) Many of the transcriptional effects of MEK inhibition are duplicative of those already identified in Loi et al, CCR 2016, and not cited. These data are not really novel except that they are in additional tumor models than those described. Can the authors take the next step and show that any of these features are critical to the activity of MEKi on T cell activity? This would be an advance.

RESPONSE:

We had already cited the 2017 paper from the same group in relation to the enhancement of MHC expression in response to MEKi and IFN γ treatment, and now included citations of the Loi et al 2016 paper in relation to the induction of gene signatures related to antigen presentation and T-cell activation, as well as to the combined anti-tumor effect of MEKi and PD-L1 blockade (see below).

We would like to note that in our PDA30364 model the MEKi single agent impact on in vivo gene signatures related to antigen presentation (Fig. 6E) is more profound than shown in Loi et al. 2016 for NCOMMS-19-16321-T Letter to Editor, page 2/14

the MMTV-Neu models (Fig. 4E in Loi et al.), and that the MEKi induced gene signature found in the 30364 model extends to various other gene sets (Fig. 6E), including multiple T-cell markers (Fig. 6F). As such, our data do – in our opinion – extend beyond those already published.

Comparison of the gene signatures reported in Loi et al. 2016 and in our study shows that treatment impact is highly dependent on the tumor model. In earlier versions of our manuscript, we had addressed these findings (for our own 3 models) quite extensively. However, we had to condense this part of the text to stay within the word count limitations and therefore decided to focus our phrasing on the impact of MEKi on the T-cell signature. Notably, this signature shows a strong correlation with the CD8+ T-cell count as detected in the tumors by means of flow cytometry (Fig. 6F; Suppl. Fig. 7B), thereby providing a direct link between MEKi action and T-cell activity (the next step, as requested by the reviewer).

This text in our original manuscript reads as follows (please note the sentence underlined):

*In the same manner as above, we found that MEKi/CD40 Ab treatment of PDA30364 tumors activates multiple pro-inflammatory pathways involved in, amongst others, immune cell infiltration, the T-cell immune response and antigen presentation (Fig. 6D-F). Interestingly, this immune signature is also induced in PDA30364 tumors treated by either of the single drugs, supporting the notion that not only anti-CD40 Ab but also MEKi **can have** clear-cut pro-inflammatory impact. The biological relevance of this observation is illustrated by the striking correlation between the strength of the T-cell signature and the CD8+ T-cell count under the different treatment conditions (Fig. 6F).*

Furthermore, we had already referred to the differential impact of MEKi on the T-cell signature in the different models:

*In view of this finding, we evaluated the T-cell signature in the context of CD8+ T-cell count across the PDA30364, B16-OVA and MC-38 models (Supplementary Fig. 7B). As expected, treatment with anti-CD40 Ab treatment induced this signature and increased CD8+ T-cell count in all three models. **In contrast**, the impact of MEKi single agent treatment ranged from activation in the PDA30364 model to suppression in the B16-OVA model, indicating that under certain conditions MEKi can suppress T-cell immunity, as suggested by our *in vitro* T-cell experiments (Supplementary Fig. 2). This differential, tumor model-dependent impact of MEKi single treatment was also observed for the other pro-inflammatory pathways that were induced in treated PDA30364 tumors (Supplementary Fig. 7C) **and is reminiscent of findings reported by Loi et al for the MMTV-neu model**³⁵.*

In view of the reviewer comments, we amended this section of the Results section as follows (see above and below; changes marked in red):

- We stated that *MEKi **can have** clear-cut pro-inflammatory impact*
- We inserted reference to the paper by Loi et al. 2016 (also in other places where we referred to MEKi/PD-1 studies!)
- We inserted a brief paragraph on the correlation between cytostatic impact of treatment and the induction of the pro-inflammatory gene signatures (see below).

*For the three tumor models examined in our study, the impact of MEKi on the immune gene signatures correlates with the single agent cytostatic effect on these tumors, as well as with the *in vitro* IC50*

NCOMMS-19-16321-T Letter to Editor, page 3/14

values. In the highly sensitive PDA30364 tumor, strong suppression of the cell cycle and biosynthesis-related pathways (Fig. 6A-B) is associated with induction of pro-inflammatory genes (Fig. 6D-F). In the least sensitive B16 tumor, marginal impact of MEKi single agent treatment on the cell cycle and biosynthesis-related gene set (Supplementary Fig. 7A) is associated with marked suppression of the immune genes (Supplementary Fig. 7B-C). For the MC38 tumor, these effects are intermediary. Taken together, these findings suggest that MEKi treatment is immunosuppressive unless accompanied by tumor cell death, an event known to be pro-immunogenic^{13,14}. Importantly, in the less sensitive B16 and MC38 tumors, MEKi-induced immunosuppression can be overcome by combining MEK inhibition with anti-CD40 Ab, resulting in synergistic anti-tumor efficacy (Fig. 3A-B), strong suppression of the cell cycle and biosynthesis-related pathways as well as induction of the pro-inflammatory gene sets (Supplementary Fig. 7).

In view of these changes, we deleted the final paragraph of this Results Section, which has become redundant:

In conclusion, MEK inhibition can induce pro-immunogenic pathways in tumors that are associated with T-cell immunity and immune cell infiltration in a manner that is closely associated with the anti-tumor impact of the treatment. Furthermore, MEKi treatment induces tumor-cell intrinsic interferon signaling and, when combined with IFN γ , can strongly enhance the upregulation of MHC class I and II levels at the cell surface of tumor cells.

Finally, we added a sentence to the 3rd paragraph of the Discussion (marked in RED):

Comprehensive analysis of the immune infiltrate in tumor samples in the context of treatment, as performed in our mouse models, presents a challenge in the clinical setting due to the need for fresh tissue samples. In view of this, we explored the use of whole tumor transcriptome analysis in the three tumor models tested, which resulted in the identification of gene signatures that mirror the mechanism of action of the drugs, that reflect the differences in anti-tumor efficacy of the treatments in the three tumor models, and that correlate with the changes in immune infiltrate as observed by flow cytometry. For instance, the extent of suppression of the cell cycle-related pathways by MEKi treatment in the B16-OVA, MC-38 and PDA30364 tumors shows an excellent correlation with the low, moderate and strong impact of MEK inhibition on the growth of these tumors, respectively. Furthermore, the induction of pro-inflammatory gene signatures is only seen in conjunction with significant anti-tumor efficacy of the treatment regimen, in particular in all three tumors subjected to MEKi/CD40 Ab treatment as well as in PDA30364 tumors treated with MEKi only. Therefore, these signatures provide suitable surrogate endpoints for the evaluation of treatment efficacy, as well as PD biomarkers for monitoring drug exposure and impact in the tumor.

2) in Figure 3, PD-1 or L1 should be shown as a comparator - the immunogenic effects of MEK inhibitors are now established, in combination with PD-1/L1, so - is this combination better or just an additional possible option - either is fine, but the relative levels of activity should be described

RESPONSE:

We added the requested experiment in which we compared the efficacy of CD40 Ab and PD-1 Ab, as single agent and in conjunction with MEKi, in the PDA30364 model, including comparison of treatment impact on the tumor immune infiltrate. While doing so, we faced the challenge of maintaining coherence and avoiding excessive expansion of our manuscript. For comparison, also the Dushyanthen et al. 2017 paper does not provide a back to back comparison of anti-4-1BB/OX40 Abs with PD-L1/PD-1 blockade as reported in Loi et al. 2016, presumably for the same reasons.

The inserted text in resp. the 3rd, 4th and 5th sections of the Results and in the Discussion read as follows (see marked copy of revised manuscript for proper context):

Results 3rd section:

Others recently reported that MEK inhibition promoted T-cell and anti-tumor activity in combination with PD-L1/PD-1 checkpoint blockade in the AT3ova, MMTV-neu and CT26 tumor models^{32,35}. Evaluation of anti-PD-1 Ab treatment in our PDA30364 model showed no single agent activity, in contrast to what is reported for the aforementioned models. Furthermore, the combination of GDC-0632 with anti-PD-1 Abs suppressed PDA30364 tumor outgrowth with comparable efficiency as treatment with GDC-0632 only (Fig. 3E, Supplementary Fig. 5A). Thus, at least in this model, the combination of MEKi with anti-CD40 Ab results in superior anti-tumor efficacy.

Results 4th section:

In comparison, treatment of PDA30364 tumors with anti-PD-1 Abs elicited only a minor increase in the CD8+ T-cell fraction and CD8+/Treg ratio (Supplementary Fig. 5B). A more profound impact on the CD8+/Treg ratio is seen under MEKi/PD-1 Ab treatment, but this is most likely due to the aforementioned suppression of CD4+ Tregs by GDC-0632. Furthermore, the cytokine production by freshly isolated tumor-infiltrating T-cells is equally enhanced for MEKi/CD40 Ab and MEKi/PD-1-treated tumors (Supplementary Fig. 5C). Altogether, these data indicate that the MEKi/CD40 Ab regimen may be more effective, because it increases both the activity of tumor infiltrating T-cells and the magnitude of the T-cell response (see further below).

Results 5th section:

Notably, a similar shift in the M1/M2 ratio was not observed in PDA30364 tumors treated with the MEKi/PD-1 Ab combination (Supplementary Fig. 5D), which could further explain the superiority of the MEKi/CD40 Ab regimen in this model.

Discussion:

Based on our experiments in the PDA30364 model, agonist anti-CD40 Abs are more effective than PD-1 blocking Abs in this respect, most likely because of their more profound impact on the immune cell infiltrate in the tumor.

Minor:

3) There are some inaccuracies as to the state of the field in the introduction. The authors cite the limited utility of immune checkpoint blockade in the opening statement of the introduction. The statement is a bit misleading. ICB has been approved in a vast variety of tumor types, not just lung and melanoma. This is a strange tennet since the authors do not test ICB. Later, "while immune checkpoints appear primarily suitable for mobilizing existing T cell responses..." - this statement reflects what is known about PD-1/L1, but not CTLA-4.

RESPONSE:

We agree with the reviewer that our opening statement provided insufficient credit to the clinical efficacy of currently available checkpoint inhibitors, and have amended our phrasing accordingly. Our opening statement now reads as follows (changes marked in RED):

First paragraph:

*Cancer immunotherapy is beginning to realize its potential in the clinic **with** immune checkpoint blockade (ICB) **eliciting** durable responses in patients with immunogenic cancers such as melanoma and lung cancer. **Nevertheless**, sub-groups of patients with these indications do not respond to ICB, and the same applies to patients with other cancer types. A major hurdle in this respect is the lower intrinsic immunogenicity of ICB-resistant tumors,*

Second paragraph:

*In order to build on the first promising results of ICB, there is a clear need to explore additional drugs and treatment regimens in clinical trials. Agonist immunostimulatory antibodies (IS-Abs) targeting activatory receptors on immune cells are a potential alternative for immune checkpoint inhibitors and have demonstrated notable results in pre-clinical models. An attractive aspect of agonist anti-CD40 Abs, which make these complementary to checkpoint inhibitors, is the capacity to enhance the priming of T-cell responses through the activation of dendritic cells (DCs). **Anti-CD40 Abs may therefore be able to turn 'cold' tumors into 'hot' tumors.** Furthermore, anti-CD40 Abs were found to make tumors more permissive*

In addition, in response to comment #2 by this reviewer, we added data providing a comparison of the impact of MEKi/CD40 and MEKi/anti-PD-L1 treatments in our pancreatic cancer model (see above).

4) The drug name GDC-0623 is erroneously reported in Figure 1 legend. (0632)

RESPONSE:

Corrected

5) Figure 2A is missing a control - what about OVA without CD40?

RESPONSE:

These controls were already included in the initial manuscript version as Suppl. Fig. 2A (Suppl. Fig. 3A in revised manuscript). In order to make this referral to these data more explicit, we clarified the text in the Results section as follows (changes marked in RED):

The MEK/ERK pathway was reported to be essential for T-cell priming and effector function. We initially examined GDC-0623 for potential suppressive impact on T-cell function in in vitro experiments with antigen-stimulated TCR-transgenic, chicken ovalbumin (OVA)-specific CD8+ OT-I T-cells. The resulting data pointed at profound inhibition of T-cell proliferation and cytokine production at tumor-effective concentrations (Supplementary Fig. 2). Similar observations were made for the other three SMi (data not shown). In contrast, testing of GDC-0623 in in vivo immunization experiments with OT-I T-cells revealed that antigen-induced T-cell priming and proliferation were only slightly affected by drug doses known to suppress tumor growth in xenograft models (Fig. 2A-B). These in vivo experiments involved systemic immunization of mice with OVA protein in combination with a systemic administration of agonist anti-CD40 Ab (anti-mouse CD40 Ab 3/23 mouse IgG1). In this setting, effective activation of the OT-1 T-cells depends on the co-administration of the anti-CD40 Ab (Supplementary Fig. 3A), which induces the activation of DCs and thereby mediates the immunogenic presentation of the OVA-derived epitope SIINFEKL to T-cells^{33,34}.

For further clarification of this aspect we did make the following amendments in our manuscript (see also our response to comment #4 by Reviewer 2):

- We added to the legend of Suppl. Fig. 3A a statement that the lack of in vivo OT-1 T-cell expansion after administration of either OVA or anti-CD40 Ab only, as shown in this figure, was verified in multiple experiments and is in line with the published data from Glennie and colleagues, from whom we obtained the anti-CD40 Ab and adapted the assay.
- Similar references were added to the relevant paragraph in the Materials & Methods section.

Notably, due to several changes in the manuscript the numbering of the supplemental figures has changed considerably.

6) Details on figure 2C are missing - what are the target cells? (also spelled typo – target T-cell on page 9

RESPONSE:

Full details on the in vivo OT-1 T-cell killing assays are provided in the Materials and Methods section. We now provided further detail on the targets cells in the legend text of Fig 2C by amending this as follows:

Three days after immunization, 2 million CFSE labeled peptide pulsed target cells (CD45.2+ splenocytes pulsed with 10 µg/ml SIINFEKL; 0.3 µM CFSE) admixed with control cells (CD45.2+ splenocytes pulsed with 10 µg/ml p53 control peptide AIYKKSQHM; 5 µM CFSE) were transferred intravenously and spleens analyzed for specific killing after 24h.

Typo was corrected

Reviewer 2 (Remarks to the Author):

In this study, Baumann et al reported a synergy between MEK inhibitors and agonist anti-CD40 immunostimulatory antibodies. Based on literature and multiple in vitro cancer cell line models, the authors selected a MEKi GDC-0623 as an optimal cytostatic small molecule inhibitor for immunoncology applications. While GDC-0623 inhibited T-cell proliferation and cytokine production in vitro, at tumor-effective dosages GDC-0623 did not significantly impair DC and T-cell functions in the in vivo OT-I T-cell immunization assays nor in the in vivo OT-I T-cell killing assays. GDC-0623 and agonist anti-CD40 Ab showed synergistic anti-tumor efficacy in three syngeneic tumor models, B16-OVA (melanoma), MC-38 (colorectal cancer), and PDA30364 (pancreatic cancer). The authors showed that in the PDA30364 model, the synergy was T-cell dependent. In addition, GDC-0623 was more toxic for the immunosuppressive M2 macrophages and MDSCs than the pro-inflammatory M1 macrophages. To understand the mechanism of the synergy, the authors performed RNAseq analyses for GDC-0623/agonist anti-CD40 Ab-treated tumor samples. Based on differential mRNA expression levels, cell cycle and biosynthesis-related pathways were down-regulated by GDC-0623, and T cell markers were up-regulated by agonist anti-CD40 Ab in all three syngeneic models. Furthermore, using the pancreatic cancer PDA30364 syngeneic tumor model, the authors also developed gene signatures for GDC-0623, agonist anti-CD40 Ab, and their combination, that correlated with treatment efficacy. These findings are significant and important for developing effective therapeutic strategies for MAPK pathway-dependent cancers. The manuscript is well written. Considering the following suggestions may help strengthen the proposed model and the manuscript.

Major comments:

1. The author indicated that the synergy of MEKi and agonist anti-CD40 Ab in tumor therapy was “T-cell mediated”. However, among the three syngeneic tumor models used in this study, only in the pancreatic cancer PDA30364 model MEKi enhanced T cell proliferation and activation as a single agent, and showed synergy with anti-CD40 Ab in T cell activation. In the MC-38 model, MEKi did not show significant effect on T cell proliferation nor synergy with anti-CD40 Ab on T cell proliferation (supplementary Fig. 5B). In the B16-OVA model, MEKi clearly suppressed T cell proliferation and showed antagonism with agonist anti-CD40 Ab in T cell proliferation (supplementary Fig. 5B). In addition, MEKi decreased the expression of many genes involved in immune cell infiltration, T-cell immune response and antigen presentation in both B16-OVA and MC-38 tumors (supplementary Fig.5C). Therefore, mechanisms other than T cell proliferation and activation may be involved in the synergy of MEKi and agonist anti-CD40 Ab in the MC-38 and B16-OVA models. Based on these results, I would recommend to remove “T-cell mediated” from the manuscript title.

RESPONSE:

We agree with the reviewer that the impact of MEKi, anti-CD40 Ab and their combination on the three tumor models is complex (see also our response to comment #1B by Reviewer 1), and that besides T-cell immunity other mechanisms play a role. In view of this, we amended the title of our manuscript as requested:

Pro-immunogenic impact of MEK inhibition synergizes with agonist anti-CD40 immunostimulatory antibodies in tumor therapy

2. At the beginning of the Results section, the authors determined GDC-0623 as the optimal cytostatic compound for immune-oncology application. However, the assay (CellTiter-Glo assay) they used to test the compounds was an ATP-based assay, which could not distinguish between cytostasis and cytotoxicity. In fact, GDC-0623 was cytotoxic for M2 macrophages, as indicated by the substantial increase in the sub-G1 population in Fig. 5C. The author should perform a viability assay, such as trypan-blue staining or PI-cell cycle analysis, to confirm the cytostatic feature of GDC-0623 in their cancer cell models.

RESPONSE:

Experimental data concerning the (balance between) cytostatic and cytotoxic impact of GDC-0632 on the three tumor models were originally provided by us as supplemental figures (Supplementary Fig. 3B-C). The request by this Reviewer for these experiments implies that the in vitro drug testing experiments should preferably be presented in a more coherent format, which is why we decided to move these data to revised Figure 1 (Fig. 1D-E). These data show that the degree of cell death induced by GDC-0632 was highest for PDA30364. The text of the Results Section was amended accordingly. For details on these and further changes to Figure 1 and the corresponding text, please see our response to comment #8 below, as well as our response to comment #1A by Reviewer 1.

3. In Fig. 1c, the authors tested the sensitivity of primary patient-derived PDA cell lines to MEKi GDC-0623 to support the point that this inhibitor has superior efficacy in KRAS-driven tumors. For that, the authors should provide the KRAS mutation status of these patient-derived PDAC cell lines.

RESPONSE:

These PDX models indeed all harbored activating mutations in K-ras. Information on the mutation status of these tumors has been added to Supplementary Table 1 and the text of the Results Section was amended accordingly (see our response to comment #8).

4. In all data in Fig.2, negative controls without OVA and anti-CD40 Ab injection should be included.

RESPONSE:

These controls were already included in the initial manuscript version as Suppl. Fig. 2A (Supplementary Fig. 3A in revised manuscript). In order to make this referral to these data more explicit, we clarified the text in the Results section as follows (changes marked in RED):

The MEK/ERK pathway was reported to be essential for T-cell priming and effector function. We initially examined GDC-0623 for potential suppressive impact on T-cell function in in vitro experiments with antigen-stimulated TCR-transgenic, chicken ovalbumin (OVA)-specific CD8+ OT-I T-cells. The resulting data pointed at profound inhibition of T-cell proliferation and cytokine production at tumor-effective concentrations (Supplementary Fig. 2). Similar observations were made for the other three SMi (data not shown). In contrast, testing of GDC-0623 in in vivo immunization experiments with OT-I T-cells revealed that antigen-induced T-cell priming and proliferation were only slightly affected by drug doses known to suppress tumor growth in xenograft models (Fig. 2A-B). These in vivo experiments involved

systemic immunization of mice with OVA protein in combination with a systemic administration of agonist anti-CD40 Ab (anti-mouse CD40 Ab 3/23 mouse IgG1). In this setting, effective activation of the OT-1 T-cells depends on the co-administration of the anti-CD40 Ab (Supplementary Fig. 3A), which induces the activation of DCs and thereby mediates the immunogenic presentation of the OVA-derived epitope SIINFEKL to T-cells^{33,34}.

For further clarification of this aspect we did make the following amendments in our manuscript (see also our response to comment #5 by Reviewer 1):

- We added to the legend of Suppl. Fig. 3A a statement that the lack of in vivo OT-1 T-cell expansion after administration of either OVA or anti-CD40 Ab only, as shown in this figure, was verified in multiple experiments and is in line with the published data from Glennie and colleagues, from whom we obtained the anti-CD40 Ab and adapted the assay.
- Similar references were added to the relevant paragraph in the Materials & Methods section.

Please note that, due to several changes in the manuscript, the numbering of the supplemental figures has changed considerably.

5. Data in Fig. 3E showed that either CD4+ T cells or CD8+ T cells were sufficient for the synergy of MEKi and anti-CD40 Ab in the PDA30364 model. Depletion of either population did not significantly affect the synergy. This observation is further supported by the effect of anti-CD40 on activating both CD4+ T cells and CD8+ T cells (Fig. 4E). This is a very interesting observation that should be further discussed.

RESPONSE:

The importance of T-cell help in the anti-tumor immune response has been a major focus of the prior work of the senior author (Offringa) of the present paper and is indeed of great relevance for the development of effective immune oncology strategies. While avoiding going overboard on this subject in the context of the present paper, we enhanced the relevant paragraph in the Discussion section as follows:

The implication of this finding is that both CD8+ and CD4+ T-cells are involved in the T-cell attack against the tumor, as we demonstrated for the PDA30364 model. This does not only reduce the risk of immune escape by downregulation/loss of MHC class I-restricted antigen presentation, as commonly found in various human and experimental tumors, including PDA^{4,44}, but also increases the repertoire of potential target antigens that can trigger T-cell-mediated tumor recognition and destruction. Of interest in this respect is that a major fraction of the neo-epitopes encoded by the tumor mutanome may be presented in the context of MHC class II²⁰. Especially for cancer types with lower numbers of somatic mutations, such as PDA, co-expression of MHC class I and II may therefore result in an increase in tumor cell immunogenicity. Moreover, the induction of CD4+ T-helper responses against tumor antigens is essential for sustained CD8+ T-cell responses, can neutralize pre-existing CD4+ T-regulatory cells as well as orchestrate activation and immune attack by innate immune cells⁴⁵⁻⁴⁷.

6. The same CD8+ T cell data were shown twice in the manuscript (Fig. 4C and Fig. 6F/supplementary Fig. 5B). I recommend to remove the CD8+ T cell data from Fig. 4C and keep them in Fig.6F and supplementary Fig. 5B.

RESPONSE:

The display of these data was indeed a dilemma for us. We wish to make clear in Fig. 4 that the change in CD8+/Treg ratio is the combined result of the increase in the CD8+ T-cell count (Fig. 4C) and the decrease in Treg count (Fig. 4D). Furthermore, a key point of Fig. 6F and Suppl. Fig. 7B (formerly Suppl. Fig. 5B) is to demonstrate the correspondence between T-cell gene signature level and CD8+ T-cell count (see comment #1B by Reviewer 1 concerning the link between MEKi-induced transcriptional changes and T-cell activity!).

If we would remove this data from Fig. 4C, we would have to refer to the later figures, which would collide with journal policies typically requesting the numbering of figures in order of presentation.

Alternatively, we could delete these panels from Fig. 6F and Suppl. Fig. 7B, but in that case we would need to cross-reference between these figures and Fig. 4C, which undoubtedly would not promote the clarity of our manuscript on this important point.

Notably, the overlap between Fig. 4C and the other two figures merely concerns the CD8+ T-cell panels. We therefore respectfully request a compromise where we mention explicitly in the legends of Fig. 6F and Suppl. Fig. 7B that said panels are the same as those in Fig. 4C:

Please note that for the sake of clarity, and to avoid cross-referencing between different figures, we duplicated the presentation of the CD8+ T-cell data from Fig. 4C in this figure.

Notably, due to several changes in the manuscript, the numbering of the supplemental figures has changed considerably.

7. In Fig.6I and supplementary Fig. 5D, the authors showed that, in cell culture models, MEKi GDC-0623 and IFN γ synergistically up-regulated MHC expression at the cell surface. This could be an important mechanism of the synergy of MEKi and anti-CD40 Ab in in vivo syngeneic tumor models. Isogenic IFN γ receptor deletion tumor models could be used to determine the role of IFN γ in mediating the synergy in vivo.

RESPONSE:

We are in the process of generating CRISPR/cas IFN γ R-ko variants of the PDA30364 cell line, as part of our follow up studies to examine MEKi mechanism of action in greater detail. Given the time it will take to carefully complete these follow up experiments, we fear that the resulting delay would significantly affect the priority of our current work, which already comprises an extensive data set.

We considered referring to published work by others in which a similar link between BRAF-inhibition and signaling by IFN γ has been demonstrated [Ho et al. Cancer Res 2014; 74, 3205]. However, this study focuses on the role of the IFN γ pathway on the myeloid immune infiltrate in the tumor. It is conceivable that this mechanism is also relevant in our models, as well as in the models presented in Loi et al 2016. However, we would like to refrain from too much speculation in this respect.

8. In the proposed model in Fig.8, the authors indicated KRASG12D tumors. However, among the three syngeneic tumor models used in the study, only PDA30364 carries KRASG12D. The authors should revise the model.

RESPONSE:

We agree with this point, and removed 'Kras-G12D' from the figure and replaced it by 'pERK'. The latter is in line with newly added data in Fig. 1C showing that all three tumor cells express high pERK levels that are profoundly inhibited by MEKi (revised Fig. 1).

In addition, we added a mutational analysis of the three syngeneic murine tumor models (Supplementary Table I), showing that mutated K-ras is the key driver mutation in PDA30364, while MC38 and B16 do not harbor K-ras mutations, but instead a number of other potential driver mutations that could stimulate the MEK/ERK pathway, either directly or through cross-talk between pathways. The text of the Results Section was amended accordingly and reads as follows (changes marked in RED: notably, these also include changes in relation to comments #2 and #3):

GDC-0623 was reported to have superior efficacy in KRAS-driven tumors, due to its capacity to block MEK feedback phosphorylation by wild type RAF. This was confirmed by the in vitro sensitivity of primary patient-derived PDA cell lines to this drug (Supplementary Fig. 1F) and is in line with the detection of activating K-ras mutations in these cell lines (Supplementary Table 1). Mutational analysis of the three mouse tumor lines revealed that, as expected, mutated K-ras is a key driver mutation in PDA30364. B16 and MC38 do not harbor K-ras mutations, but instead carry a number of other potential driver mutations that could stimulate the MEK/ERK pathway, either directly or through crosstalk between signaling pathways (Supplementary Table 1). Notably, all three tumor cell lines express elevated pERK levels that are profoundly inhibited through incubation with GDC-0623 (Fig. 1C). Nevertheless, the impact of MEK inhibition on cell viability is most prominent in PDA30364, in line with the pivotal role of mutated K-ras in this cell line (Fig. 1A). Detailed analysis of the balance between cell

death and G0/G1 arrest, as induced in the three tumor cell lines by GDC-0623, confirmed that the degree of cell death was highest for PDA30364, whereas this drug induced a blend of cell death and stasis in B16 and MC38 (Fig. 1D-E).

Minor comments:

9. In the histogram plots in Fig.2 and supplementary Fig. 2, the author should indicate what different color curves are for.

RESPONSE:

Amended as requested; please note that suppl. Fig 2 is now suppl. Fig. 3

10. Dosages and concentrations of agents used in experiments should be included in the figure legends.

RESPONSE:

Amended as requested.

11. Significance needs to be added to Fig. 6I and supplementary Fig. 5D.

RESPONSE:

Amended as requested; please note that suppl. Fig 5D is now suppl. Fig. 7D

REVIEWERS' COMMENTS:

Reviewer #1 (Remarks to the Author):

The additional changes and experiments are sufficient

Reviewer #2 (Remarks to the Author):

In this revised version, the authors have answered my previous questions, and have substantially improved the quality of the manuscript. I recommend the manuscript to be published in Nature Communications.

Reviewer #3 (Remarks to the Author):

The exome sequencing was performed as what was done by most labs worldwide, and is correct. Most of the exome sequencing data processing steps were also reasonable. However, for statements on somatic mutation, authors should make it clearer. MuText v1 can only detect somatic point mutation, and MuTect2 can detect both somatic point mutation and somatic indels. How the results from the two tools were combined should be stated. Somatic gene deletion detection is not a trivial task. Authors should state concisely the detection principle behind the custom script, or provide a web link where the script can be found (such as Github).

Authors wrote:

"In case of PDA30364, the tumor cell lines and spleen of the original mouse was used as a germline control."

It's confusing as the tumor cell lines PDA30364 should be the samples somatic mutations are called from, and only the spleen of the original mouse should be treated as germline control. Also, in the Supplementary Table 1, only spleen of the original mouse was stated as germline control.

Authors wrote:

"Categorization of MAPK cascade-associated genes."

This sentence is not complete. Maybe changing it to "For the categorization of MAPK cascade-associated genes, " will make the expression clearer.

Point by point response to reviewer #3

Reviewer #3 (Remarks to the Author):

1. The exome sequencing was performed as what was done by most labs worldwide, and is correct. Most of the exome sequencing data processing steps were also reasonable. However, for statements on somatic mutation, authors should make it clearer. MuText v1 can only detect somatic point mutation, and MuTect2 can detect both somatic point mutation and somatic indels. How the results from the two tools were combined should be stated. Somatic gene deletion detection is not a trivial task. Authors should state concisely the detection principle behind the custom script, or provide a web link where the script can be found (such as Github).

We corrected the reference to this methodology in the Materials and Methods section. In particular:

- *We referred to MuTECT2 only*
- *We provided details on our custom script in Supplementary Figure 11*

2. Authors wrote:

"In case of PDA30364, the tumor cell lines and spleen of the original mouse was used as a germline control."

It's confusing as the tumor cell lines PDA30364 should be the samples somatic mutations are called from, and only the spleen of the original mouse should be treated as germline control. Also, in the Supplementary Table 1, only spleen of the original mouse was stated as germline control.

Corrected: we now only refer to the normal spleen as germline control

3. Authors wrote:

"Categorization of MAPK cascade-associated genes."

This sentence is not complete. Maybe changing it to "For the categorization of MAPK cascade-associated genes, " will make the expression clearer.

Thank you for spotting this. Corrected: we amended this sentence accordingly